# Dynamic proteomic and phosphoproteomic atlas of corticostriatal axons in neurodevelopment

**Vasin Dumrongprechachan[1,2], Ryan B Salisbury[3], Lindsey Butler[1], Matthew L MacDonald[3]\*, Yevgenia Kozorovitskiy[1,2]\***

[1]Department of Neurobiology, Northwestern University, Evanston, United States; [2]The Chemistry of Life Processes Institute, Northwestern University, Evanston, United States; [3]Department of Psychiatry, University of Pittsburgh, Pittsburgh, United States

**Abstract** Mammalian axonal development begins in embryonic stages and continues postnatally. After birth, axonal proteomic landscape changes rapidly, coordinated by transcription, protein turnover, and post-translational modifications. Comprehensive profiling of axonal proteomes across neurodevelopment is limited, with most studies lacking cell-type and neural circuit specificity, resulting in substantial information loss. We create a Cre-dependent APEX2 reporter mouse line and map cell-type-specific proteome of corticostriatal projections across postnatal development. We synthesize analysis frameworks to define temporal patterns of axonal proteome and phosphoproteome, identifying co-regulated proteins and phosphorylations associated with genetic risk for human brain disorders. We discover proline-directed kinases as major developmental regulators. APEX2 transgenic reporter proximity labeling offers flexible strategies for subcellular proteomics with cell type specificity in early neurodevelopment, a critical period for neuropsychiatric disease.

**\*For correspondence:**
macdonaldml@upmc.edu
(MLMacD);
Yevgenia.Kozorovitskiy@
northwestern.edu (YK)

**Competing interest:** The authors declare that no competing interests exist.

## Editor's evaluation

Knowledge of the protein composition of defined subcellular compartments is of key importance for the characterization of protein machines that mediate defined cellular functionalities. The current paper presents a novel mouse line that will serve as a tool of fundamental value in this context – a Cre-inducible APEX2 reporter mouse line for acute ex-vivo proximity biotinylation. The authors provide compelling evidence documenting the usefulness of the novel reporter line, describing circuit-specific proteomes and phosphoproteomes in the corticostriatal system of the mouse brain during development. The biological insights deduced from bioinformatic analyses of the proteomic data are convincing. The new APEX2 reporter mouse line will be of substantial interest to researchers in many fields of mammalian biology.

## Introduction

Neurons are morphologically diverse cells that compartmentalize cellular signaling and information processing in cell bodies, dendrites, and axons, enabling spatiotemporal control over synaptic transmission. While these subcellular domains are maintained and regulated by both local and distal cues, local proteome regulation is particularly important for fine-tuning neural connectivity during circuit development and in the context of plasticity (*Gonzalez-Lozano et al., 2016*; *Poulopoulos et al., 2019*; *Schanzenbächer et al., 2018*). During the embryonic and early postnatal stages, axons rapidly grow and navigate to target regions across the brain (*Winnubst et al., 2019*). Developing axons require local protein synthesis and dynamics to coordinate axon outgrowth and growth cone collapse

(*Lin and Holt, 2007*). Although axons contain machinery for local protein synthesis (*Hafner et al., 2019*), a large fraction of the neuroproteome is synthesized in the soma and trafficked throughout the cell, as evidenced by generally greater abundance of transcripts in the soma, compared to neurites (*Glock et al., 2021*). Thus, proteomic rather than transcriptomic techniques are necessary to capture the functional state of axons; this is especially important for axon guidance signaling that is highly regulated by post translational modifications such as protein phosphorylation (*Costa-Mattioli et al., 2009*). In addition, neuronal cell types are characterized by distinct patterns of gene expression (*Saunders et al., 2018*; *Zeisel et al., 2018*), implying biodiversity of their somatic, dendritic, and axonal proteomes. Therefore, it is important that measurements of the microdomain proteomes also contain cell-type specific information.

The postnatal proteome of developing axons remains largely unknown for any neuronal type. Mapping axonal proteomes from specific neurons directly in the vertebrate brain remains extremely challenging. Axonal compartments, such as axon growth cones, are typically isolated by tissue dissociation and sorting techniques, resulting in substantial losses of axon components (*Chauhan et al., 2020*; *Poulopoulos et al., 2019*). Axons from different circuits and cell types are intermingled, and isolation techniques that lack genetic targeting likely mask neuronal class specific features. Several genetically encoded proteomics strategies have been developed to profile cell-type-specific proteomes in the mouse brain, including metabolic labeling (*Alvarez-Castelao et al., 2017*; *Krogager et al., 2018*) and proximity labeling strategies (*Dumrongprechachan et al., 2021*; *Hobson et al., 2022*; *Rayaprolu et al., 2021*; *Spence et al., 2019*; *Uezu et al., 2016*). We and others have previously shown that APEX-based proximity labeling in the acute brain sections is highly efficient and robust, providing sufficient material with both cell-type and subcellular compartment specificity such that there is no need for tissue dissociation, biochemical fractionation, or subject pooling (*Dumrongprechachan et al., 2021*; *Hobson et al., 2022*). APEX2 was engineered to be highly efficient for rapid protein biotinylation within seconds to minutes in the presence of hydrogen peroxides (*Lam et al., 2015*), providing superior labeling speed over other proximity labeling methods (*Hung et al., 2016*; *Lobingier et al., 2017*). Our prior work also demonstrates that rapid labeling speed of virally transduced APEX is capable of taking proteomic snapshots and capturing changes in the proteome within 3–4 hr time windows (*Dumrongprechachan et al., 2021*), but it is incompatible with applications that require early postnatal time points. Although APEX-based proximity labeling is a promising candidate for accessing the proteomic landscape of neurodevelopment, significant modifications to the overall strategy are needed in order to successfully map genetically targeted axonal proteomes in the mouse brain during postnatal development.

In this study, we focus on the postnatal development of corticostriatal projections. Corticostriatal systems represent a powerful model for studying axon development and axon guidance. In rodents, corticostriatal projections arise in the embryonic period and continue to innervate the striatum postnatally (*Nisenbaum et al., 1998*; *Sheth et al., 1998*; *Sohur et al., 2014*), where growth cones start to disappear around postnatal day P7, based on the reduction of growth-associated protein 43 (GAP43) immunostaining (*Dani et al., 1991*). As axon outgrowth declines after P7, the maturation of striatal neuron physiology becomes more pronounced, with high levels of dendritic spinogenesis, synaptogenesis, and tuning of intrinsic excitability until reaching maturity (P8 – P30) (*Kozorovitskiy et al., 2012*; *Kozorovitskiy et al., 2015*; *Krajeski et al., 2019*; *Lieberman et al., 2018*; *Peixoto et al., 2016*). Therefore, the dynamic phases of axonal development in the striatum raise a question about how the local axonal proteome changes across the first three postnatal weeks. Further understanding of typical neurodevelopment provides insight into how these processes break down in disease, as corticostriatal circuits are implicated in a number of neurodevelopment disorders such as autism spectrum disorder and schizophrenia. Because the striatum receives excitatory inputs from several brain regions (*Hunnicutt et al., 2016*), conventional approaches without a genetic targeting strategy cannot distinguish axons from different input types. To date, no genetically targeted proteome or phosphoproteome of axons across development has been reported. Phosphorylation state information, along with proteomics measurements, would provide an unparalleled window into local protein dynamics critical for neurodevelopment. Current APEX-based proteomics approaches primarily use Cre-dependent adeno-associated viral transduction (AAV) to deliver APEX reporters. This workflow restricts possible experimental timelines, because optimal expression of Cre-dependent AAV may take 1–3 weeks, rendering the early postnatal proteome intractable.

To address both technical limitations and biological questions in this study, we created a Cre-dependent APEX2 reporter mouse line using CRISPR knock-in. We demonstrated that the APEX reporter line can be used with multiple Cre-driver lines and is suitable for proteomics applications, including in the early postnatal period. We mapped the temporal expression patterns of the corticostriatal projection proteome during postnatal development. Combining APEX-based proximity labeling with phosphopeptide enrichment enabled a characterization of the local phosphoproteome in axons, revealing dynamic regulation of proline-directed kinases and phosphosites in corticostriatal projection during development.

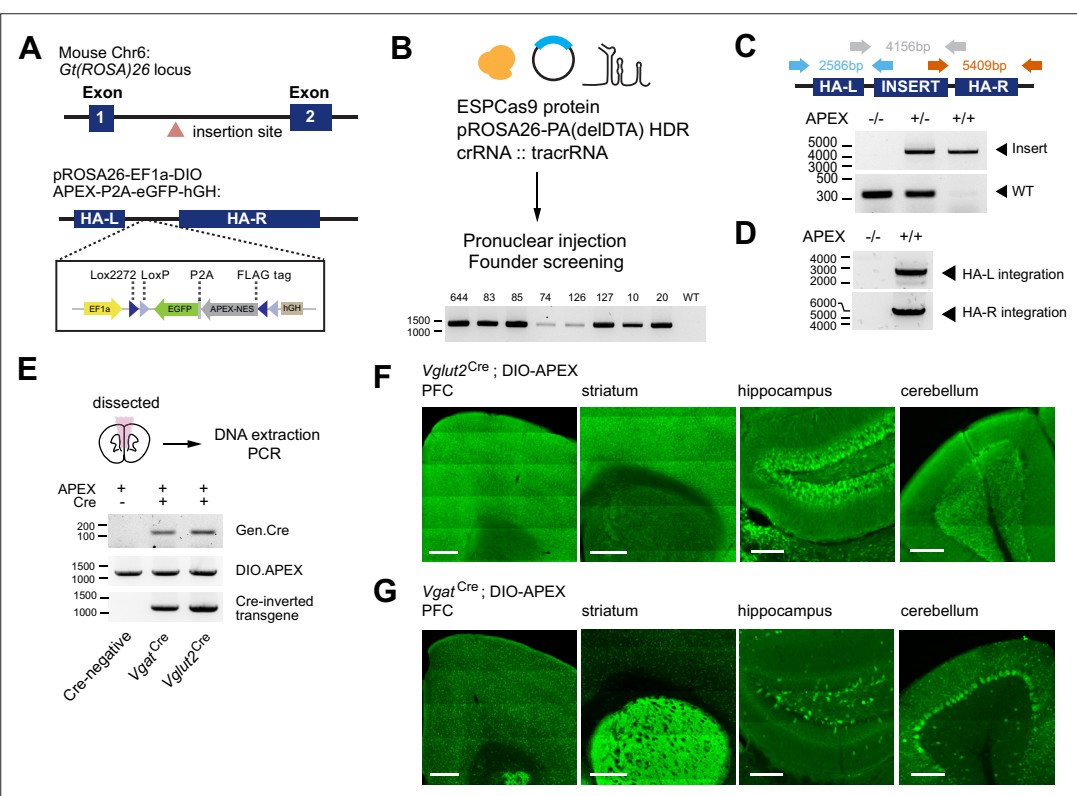

**Figure 1.** Generation of a Cre-dependent APEX2 reporter mouse line. (**A**) Design of a Cre-dependent APEX reporter mouse line. A Cre-dependent APEX transgene under the control of EF1a promoter is targeted for knock-in at the *Gt(ROSA)26* locus in the mouse genome. Inverted APEX-NES (nuclear exporting sequence) is flanked by lox sites in a double floxed inverted orientation (DIO). (**B**) CRISPR/Cas9 strategy for mouse line generation. Enhanced specificity Cas9 (ESPCas9), targeting vector, and crRNA::tracrRNA duplex were used in pronuclear injection to generate founders. Agarose gel electrophoresis shows founders containing APEX transgene. (**C**) PCR amplification across the APEX transgene. Gray arrows: 317 bp WT loci and ~4 kb APEX transgene. (**D**) PCR amplification across the homology arms evaluating genomic integration. Blue arrow, left homology arm (HA-L), 2586 bp. Red arrow, right homology arm (HA-R), 5409 bp. (**E**) PCR amplification of genomic DNA extracted from the brain for the presence of Cre, the APEX transgene before and after Cre-mediated recombination in Cre[-/-], *Vgat*[Cre+/-] and *Vglut2*[Cre+/-] animals. (**F**) Cre-dependent expression of EGFP reporter in the prefrontal cortex (PFC), striatum (STR), hippocampus (HC), and cerebellum (CB) of *Vglut2*[Cre] mice crossed to APEX reporter line (immunostaining using anti-GFP antibody). Scale bars, 500 µm for PFC and STR, and 200 µm for HC and CB. (**G**) same as (**F**) but for *Vgat*[Cre].

The online version of this article includes the following source data and figure supplement(s) for figure 1:

**Source data 1.** Raw and uncropped gel images for *Figure 1b, c, d and e*.

**Figure supplement 1.** Cas9 off-target analyses in founder animals.

**Figure supplement 1—source data 1.** Raw and uncropped gel images for *Figure 1—figure supplement 1a*.

## Results

### Generation of a Cre-dependent APEX2 reporter mouse line

We created a Cre-dependent APEX reporter mouse line for broad applications in neuroscience and biology. A monomeric APEX2 variant was chosen because of its small size and efficient activity for proximity labeling of proteins (*Lam et al., 2015*). A double-floxed inverted orientation of the APEX transgene (DIO-APEX2-P2A-EGFP) was introduced into the *Gt(ROSA)26* locus under the EF1a promoter using CRISPR/Cas9 strategy (*Figure 1a*). Zygotes were microinjected with enhanced specificity eSpCas9, HDR template and crRNA:tracrRNA duplex, using a previously validated gRNA sequence targeting the *Gt(ROSA)26* locus (*Chu et al., 2016*). We identified 8 out 112 founder mutants by genotyping PCR (*Figure 1b*, *Supplementary file 1*). We confirmed germline transmission and mendelian inheritance by crossing selected founders to C57BL/6 wildtype mice over two generations. The reporter line was bred to homozygosity by crossing F2 heterozygotes (*Figure 1c*). We evaluated transgene integration using long-range PCR amplification. Using primers inside the transgene and outside the left or right homology arms, respectively, we found that the size of PCR amplicons matched the predicted sequences, suggesting that transgene integration was site-specific (*Figure 1d*). In addition, we PCR amplified and sequenced top eight predicted off-target sites with NGG PAM across 6 founders. We found a small number of off-targets that were affected by Cas9 activity on non-coding protein regions (*Figure 1—figure supplement 1a-c*, *Figure 1*, *Supplementary file 2*). We did not observe noticeable phenotypes in either heterozygous or homozygous animals. From this point onward, we chose to work with founder line #83 (*Figure 1—figure supplement 1a-c*). To evaluate conditional expression of the APEX transgene, the APEX reporter line was crossed to *Vglut2*^Cre (*Slc17a6*^Cre) or *Vgat*^Cre (Slc32a1^Cre) mice (*Vong et al., 2011*). We performed direction-specific PCR to assess Cre-dependent inversion of the APEX transgene using genomic DNA that was directly extracted from the cortex. Inverted transgene was only detected in Cre-expressing brain tissues (*Figure 1e*), confirming Cre-dependent recombination. We also inspected the pattern of EGFP reporter expression across multiple brain regions in both *Vglut2*^Cre and *Vgat*^Cre crosses (*Figure 1f–g*). Dense reporter expression was observed in the cortex for *Vglut2*^Cre and in the striatum for *Vgat*^Cre, respectively. In the hippocampus, the granule cell layer in the dentate gyrus was clearly visible in *Vglut2*^Cre cross, with smaller numbers of GABAergic cells present in the hilus of *Vgat*^Cre cross. In the cerebellum, Purkinje cell layer was observed in *Vgat*^Cre and absent in *Vglut2*^Cre mice crossed to the APEX reporter. Both genetic and histological evidence demonstrates Cre-dependent control of the APEX transgene.

### Ex vivo biotinylation workflow for cell-type-specific proteomics

To broadly distribute APEX inside neurons, we included the nuclear exporter sequence (NES) at the C-terminus of the APEX protein. We confirmed APEX localization using anti-FLAG immunofluorescent staining (*Figure 2a*). APEX localizes in the cytosolic compartment compared to untargeted EGFP structural markers, separated by a P2A linker. To demonstrate the utility of the APEX reporter line for proteomics, we used biotin phenol (BP) and $H_2O_2$ to induce biotinylation of proteins in APEX expressing neurons in the *Vglut2*^Cre and *Vgat*^Cre crosses. We increased BP incubation time to 2 min from previously published APEX-mediated biotinylation protocols to optimize protein labeling in thick brain tissue (*Dumrongprechachan et al., 2021*), since APEX has been largely characterized and applied in cell culture (*Hung et al., 2016*; *Lobingier et al., 2017*; *Paek et al., 2017*) and small organisms (*Chen et al., 2015*; *Reinke et al., 2017*). Our prior work and other studies have demonstrated that AAV-mediated APEX expression and biotinylation in the mouse brain is sufficient for cell-type specific mass spectrometry-based (MS) proteomics (*Dumrongprechachan et al., 2021*; *Hobson et al., 2022*). To label proteins in APEX-expressing tissues, acute brain slices of 250 µm thickness were prepared from *Vglut2*^Cre;APEX animals and incubated in ACSF supplemented with 500 µM BP up to 1 hr. Following BP incubation, slices were briefly washed in ACSF and biotinylation was induced by 0.03% $H_2O_2$ treatment for 2 min and quenched with Na-ascorbate ACSF (*Figure 2b*). Streptavidin dot blot analysis of cortical protein lysates shows increasing amounts of biotinylated proteins over 60 min (*Figure 2—figure supplement 1a*). Western blot analysis of total prefrontal cortex (PFC) lysates at 60 min BP incubation shows reduced protein abundance in *Vgat*^Cre compared to *Vglut2*^Cre, with negligible labeling in the Cre-negative APEX controls (*Figure 2—figure supplement 1b*). As expected, the level of biotinylation in *Vglut2*^Cre PFC sample was greater than that of *Vgat*^Cre, consistent with a greater number of glutamatergic neurons relatively to GABAergic neurons present in the neocortex, since

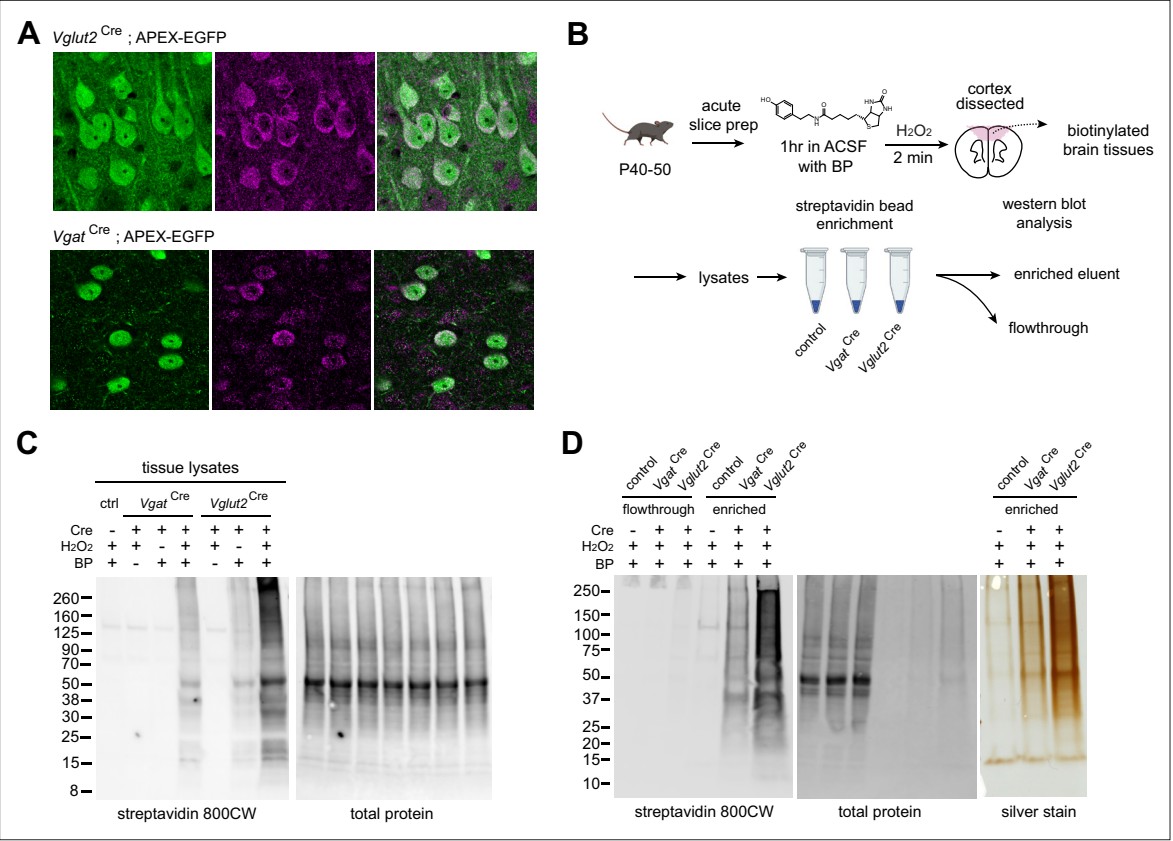

**Figure 2.** APEX-mediated biotinylation in brain tissue. (**A**) APEX subcellular localization in the mPFC of *Vglut2*[Cre];APEX and *Vgat*[Cre];APEX (immunostained EGFP 488, anti-FLAG 647 for APEX, scale bars 20 μm). (**B**) Sample preparation schematic for protein analysis. Acute slices were incubated in carbogenated artificial cerebrospinal fluid (ACSF) supplemented with 500 μM biotin phenol for 1 hr. Sections were briefly rinsed in ACSF. APEX was activated by 0.03% $H_2O_2$ ACSF for 2 min and quenched in Na ascorbate ACSF solution. Protein lysates were prepared from dissected tissues. Streptavidin beads were used to enrich tagged proteins for analysis. (**C**) Cell-type-specific biotinylation in Cre-negative, *Vgat*[Cre], and *Vglut2*[Cre] crosses. Western blot analysis of cortical protein lysates. Biotinylated proteins and total protein loading control were detected by streptavidin-CW 800 and REVERT 700 stain, respectively. (**D**) Validation of streptavidin bead enrichment. *Left*, detection of biotinylated proteins in flow-through and enriched fractions. *Right*, silver stain gel of enrichment output proteins. Differential enrichment outputs reflect differences in GABAergic and glutamatergic cell proportions in the cortex.

The online version of this article includes the following source data and figure supplement(s) for figure 2:

**Source data 1.** Raw and uncropped blots for *Figure 2c and d*.

**Figure supplement 1.** Optimization of biotin phenol ACSF incubation.

**Figure supplement 1—source data 1.** Raw and uncropped blots for *Figure 2—figure supplement 1a, b*.

under ~20% of cortical neurons are GABAergic (*Keller et al., 2018*). Using the 1 hr-long incubation protocol, we confirmed that biotinylation is both Cre and APEX-dependent (*Figure 2c*). We observed some labeling in *Vglut2*[Cre+] samples during 1 hr BP incubation without $H_2O_2$, and labeling efficiency was greatly enhanced in the presence of $H_2O_2$. To streamline sample preparation for MS-based proteomics, we evaluated the enrichment method using streptavidin magnetic beads (*Figure 2d*). Successful enrichment was verified by the depletion of streptavidin signal in flowthrough fractions and the presence of high streptavidin signal in the enrichment output, where the corresponding total amount of enriched proteins is shown by silver staining. Altogether, these experiments establish a method for selective proteome labeling, which we next apply to profile the proteome of excitatory cortical inputs to the striatum across postnatal development.

## Selective biotinylation of corticostriatal axons across postnatal development

Glutamatergic inputs into the striatum develop rapidly across embryonic and early postnatal stages (*Kuo and Liu, 2019*; *Peixoto et al., 2016*), yet deep knowledge of the overall proteome dynamics in developing corticostriatal projections has remained out of reach for technical reasons. To capture axonal proteome dynamics during this process, we crossed the *Rbp4*<sup>Cre</sup> mouse line to our APEX reporter (*Gerfen et al., 2013*). The *Rbp4*<sup>Cre</sup> mouse line expresses Cre recombinase under the control of the retinol binding protein 4 (*Rbp4*) gene promoter. A previous characterization of this line shows Cre-mediated reporter expression, enriched in the accessory olfactory bulb, layer 5 pyramidal tract neurons throughout the neocortex, and hippocampal granule cells in the dentate gyrus (*Harris et al., 2014*). As expected, *Rbp4*<sup>Cre</sup>;APEX expression is primarily restricted to layer 5 neurons which also send corticostriatal projections (*Figure 3a*). We showed that EGFP reporter expression is observed in the striatum throughout postnatal time period P5-P40 (*Figure 3a*). Western blot analysis after ex vivo biotinylation detects labeled proteins in both cortical and striatal tissues above the negative control, including at the earliest P5 time point (*Figure 3b*, *Figure 3—figure supplement 1a*). This is especially important because viral based genetically encoded proteomics is limited for early development applications (*Figure 3—figure supplement 1b*). No somatic expression was observed in the striatum; therefore, biotinylated proteins in the striatal lysates represent corticostriatal axonal proteins. Having validated axonal targeting strategy, we used mass spectrometry-based bottom-up proteomics to identify and quantify the axonal proteome of the striatum. In addition, to rigorously evaluate the extent of axonal enrichment for each protein, we included a somatic compartment control in the experiment. We used adeno-associated viral transduction to express nuclear localized H2B.APEX broadly in the cortex of *Rbp4*<sup>Cre</sup> mice. Immunofluorescence (*Figure 3—figure supplement 1c, d*) and western blot analysis (*Figure 3—figure supplement 1e, f*) show that protein biotinylation in H2B.APEX-expressing *Rpb4*<sup>Cre</sup> animals is restricted to the cortex and absent in the striatum. Thus, H2B.APEX presents one compelling option for somatic compartment control because its labeling is primarily restricted to the nucleus and soma. Although this comparison is appropriate for this study, some observed proteome differences from H2B.APEX cortex vs. *Rbp4*<sup>Cre</sup>;APEX mouse line comparison could arise from factors such as viral expression vs. *Gt(ROSA)26*-locus expression levels, or cytosolic vs. histone-fused APEX.

In our experimental design (*Figure 3c*), we biotinylated and enriched axonal proteins from the striatum across 4 time points (~P5, P11, P18, and P50, n=5 biological replicates) (*Supplementary file 3*). We targeted these early time points, because striatal circuits start to transition from axon innervation to synaptogenesis, which lead to maturation of striatal neurons as animals enter adolescence and adulthood (*Kuo and Liu, 2019*). Additionally, we included samples prepared from the *Rbp4*<sup>Cre+</sup> cortex expressing H2B.APEX as the somatic compartment control (n=3) from P18 animals. Non-labeling controls were collected from Cre-negative cortical and striatal samples that underwent the same ex vivo biotinylation procedure (n=3, 4 for cortex and striatum replicates, respectively). Enriched biotinylated proteins were digested on-bead, and peptides were barcoded with tandem mass tag (TMT) for mass spectrometry analysis. TMT-labeled peptides were mixed equally and enriched for phosphopeptides. The unbound flowthrough was fractionated using the high-pH reverse fractionation to measure protein abundance. To validate the application of flowthrough for protein abundance analysis, in a separate set of samples, we split digested TMT-labeled peptides into two equal volumes: 50% unenriched (UE) and 50% enriched for phosphopeptides (PE) to generate flowthrough (FT) for analysis (n=3 replicates) (*Supplementary file 3*). We compared three pairs of UE and FT samples at the peptide and protein levels using log2-transformed unnormalized intensities (*Figure 3—figure supplement 3a, b*). We found that UE and FT are highly correlated at the peptide level (Spearman correlation, $R>0.92$) (*Figure 3—figure supplement 3c*). Only 252 out of 13,905 peptide groups (1.8%) are significantly changed after phosphoenrichment (123 decrease, and 129 increase in intensities with q-value <0.05) (*Figure 3—figure supplement 3d*). In addition, there is no apparent relationship between the number of acidic residues and log2 FC (*Figure 3—figure supplement 3e, f*). We obtained even higher correlation between UE and FT after protein summarization ($R>0.98$) (*Figure 3—figure supplement 3g-k*). This suggests that relative protein abundance in FT is comparable to UE. Only 4 out of 2088 proteins (~0.2%) changed in abundance after phosphoenrichment. Altogether, peptide loss in the flowthrough after phosphoenrichment has limited impact on protein-level summarization in our dataset, supporting the use of this approach.

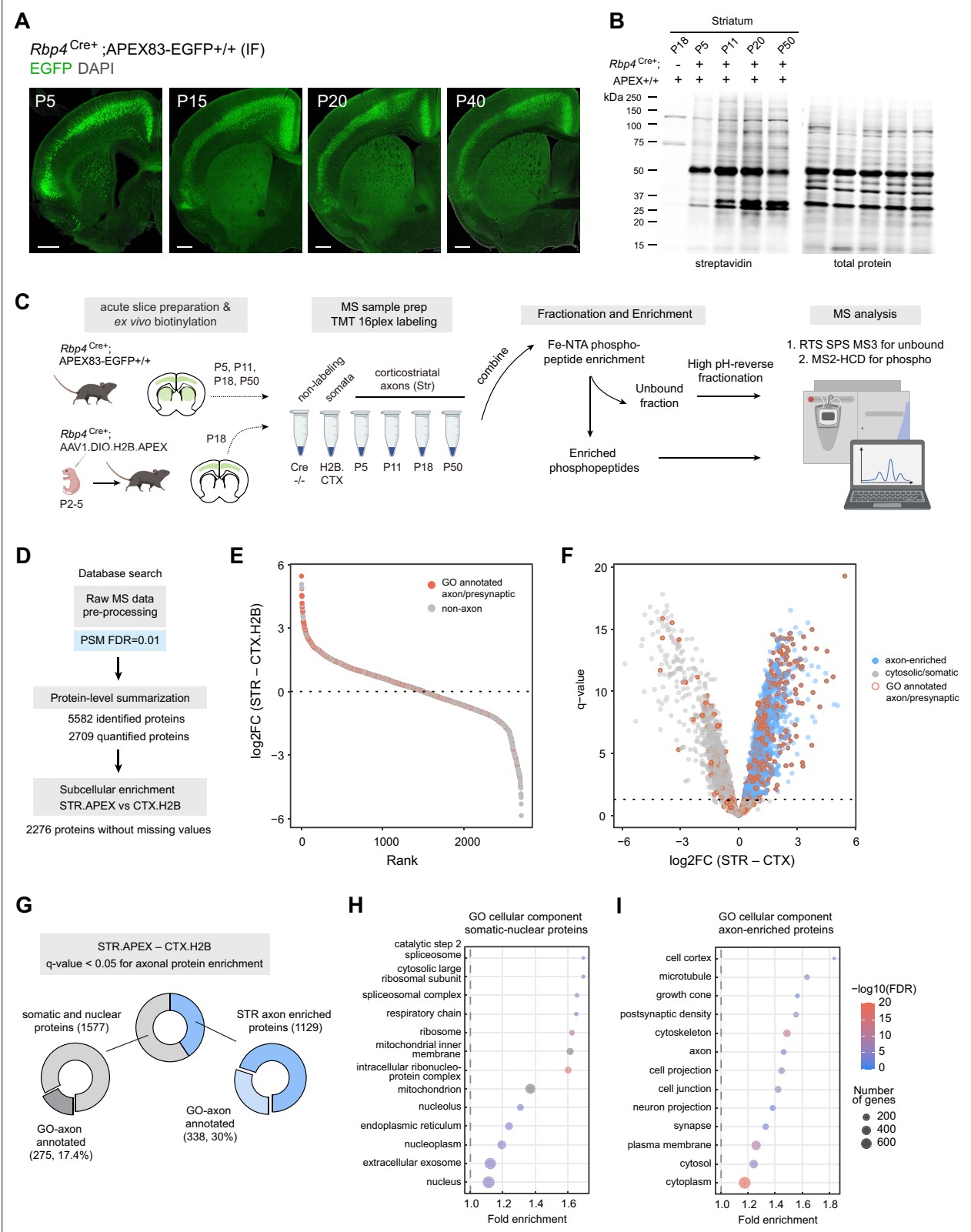

**Figure 3.** Genetic targeting approach for proteomics analysis of corticostriatal axonal compartment. (**A**) Cre-dependent APEX.EGFP expression in excitatory cortical inputs to the striatum. Expression pattern of APEX.EGFP (immunostained EGFP) in *Rbp4*[Cre+];APEX-EGFP+/+animals across postnatal days 5, 15, 20, and 40. Coronal brain section, scale bar: 1 mm. (**B**) Biotinylation of proteins in corticostriatal axons across development. Western blot analysis of striatal lysates prepared from biotinylated acute brain slices. Acute slices were prepared and biotinylated from *Rpb4*[Cre+];APEX animals at

*Figure 3 continued on next page*

*Figure 3 continued*

the indicated age. Cre-negative APEX reporter line at P18 was used as non-labeling control. *Left*, streptavidin blot. *Right*, total protein loading control. Lower amount of protein biotinylation at early age (P5) is consistent with developing innervation by axonal projections (see (**A**) ). (**C**) Sample preparation workflow for proteomics analysis. Acute slices were prepared and biotinylated from *Rbp4*^Cre+;APEX or *Rbp4*^Cre+ injected with H2B.APEX at the indicated time points. Streptavidin beads were used to enrich biotinylated proteins for the bottom-up proteomics workflow. Samples were labeled with tandem mass tags (TMT) and combined. TMT mixtures were enriched for phosphopeptide analysis using Fe-NTA tips. The unbound fractions were fractionated using high pH-reverse phase resin for protein abundance analysis. Synchronous precursor selection (SPS) MS3 method was used for proteome analysis. MS2 with high energy collisional dissociation (HCD) method was used for phosphopeptide analysis. (**D**) Data analysis flow chart. Raw MS data were analyzed by Proteome Discoverer with false discovery rate (FDR) set at 1% for peptide spectrum matches (PSMs). Protein-level summarization was performed using MSstatsTMT using unique peptides. Striatal (STR) and cortical (CTX.H2B) samples prepared from P18 were compared to determine the axon-enriched proteome. Proteins with missing values were removed prior to this comparison. (**E**) Log2-fold change (FC) plot for STR–CTX comparison. Proteins were ranked by their log2FC. Red and gray dots, proteins with or without presynaptic or axon gene ontology annotations, respectively. (**F**) Volcano plot for the STR–CTX comparison. Blue dots, axon-enriched proteins with log2FC >0 and q-value <0.05 (horizontal dotted line). Gray dots, somatic or nuclear proteins that did not pass axon enrichment cutoff. Red border, proteins with presynaptic or axon gene ontology annotations. (**G**) Proportion of proteins with GO axon/presynaptic annotation for somatic/nuclear and axon protein lists. Using STR–CTX comparison, the proteome is classified into somatic/nuclear and axon-enriched proteins. (**H**) Over-representation analysis using gene ontology (GO) cellular component database for somatic and nuclear enriched protein list. Dot plot ranked by fold enrichment. Dot size and color correspond to the number of genes, and -log10(FDR), respectively. (**I**) Same as (**H**), but for the axon-enriched protein list.

The online version of this article includes the following source data and figure supplement(s) for figure 3:

**Figure supplement 1.** Cre-dependent biotinylation of *Rbp4*^Cre+ corticostriatal projecting neurons.

**Figure supplement 1—source data 1.** Raw and uncropped blots for *Figure 3b*, and *Figure 3—figure supplement 1a, e, f*.

**Figure supplement 2.** Quality control plots for the proteomic data.

**Figure supplement 3.** Evaluation of the flowthrough fraction for protein abundance.

We adapted our previous analysis workflow using MSstatsTMT for protein summarization, and comparison with Cre-negative samples for removing non-specific binding contaminants (***Dumrong-prechachan et al., 2021***). MSstatsTMT was used to summarize peptide intensity into protein abundance (***Huang et al., 2020***). We identified 5582 proteins and quantified 2276 out of 2709 proteins without missing values across conditions (***Figure 3d Supplementary file 4***). Boxplot of normalized protein abundance shows aligned medians across samples (***Figure 3—figure supplement 2a***). Principle component analysis (PCA) shows a distinct separation of sample groups with minimal TMT plex effects (***Figure 3—figure supplement 2b***). To evaluate the extent of axonal enrichment for each protein, we used moderated t-tests implemented in MSstatsTMT to compare P18 striatal samples to the P18 H2B.APEX nuclear/somatic controls. We rank proteins by their log2 fold change (STR–CTX. H2B) (***Figure 3e***; ***Supplementary file 5***). Proteins with greater log2FC are more likely to have prior axon/presynaptic gene ontology annotations, consistent with a greater GO-axon term annotation rate as rank increases (***Figure 3—figure supplement 2c***). Proteins with q-value <0.05 and log2FC >0 were considered axonally enriched, depicted in the volcano plot (***Figure 3f***). Using this criterion, we loosely classified axonal proteins in the dataset into 1129 axon-enriched proteins and 1577 somato-nuclear proteins, with 30% and 17.4% having GO-axon annotations, respectively (***Figure 3g***). Overrepresentation analysis using the cellular component GO database shows an enrichment of cell projection-related terms for the axon-enriched proteome (*e.g.* axon, neuron projection, synapse) and nucleus and mitochondrion-related terms for the somato-nuclear proteome (*e.g.*, cytosolic ribosomal subunits, mitochondria, nucleoplasm) (***Figure 3h–i***; ***Supplementary file 6***). In addition, we also found postsynaptic density proteins such as PSD95 in our axon-enriched proteome. Together, rank based analysis confirms that our approach successfully enriched axonal proteome and GO analysis breaks down the overall subcellular compartments present in the axons. This demonstrates a workflow for mapping axonal proteomes using the Cre-dependent APEX reporter mouse line.

## Temporal trajectories of the corticostriatal axonal proteome

Next, we evaluate how the axonal proteome changes across postnatal development. We stringently include only the proteins that were quantified in all biological replicates in this analysis. First, we globally assess whether protein is significantly altered across time, using the least-squares linear regression implemented in the maSigPro R package (***Conesa et al., 2006***). Neonate (P5) was chosen as the reference time point. maSigPro reported 2274 proteins with corresponding adjusted p-values

(*Supplementary file 7*). As expected, the majority of the detected proteome was highly dynamic, with 1981 out of 2274 proteins significantly changing (adjusted p-values < 0.05). To better observe protein expression patterns, we performed hierarchical clustering of protein abundance, based on time points, revealing 8 general developmental trajectories (*Supplementary file 7*). The heatmap (*Figure 4a*) shows individual protein abundance normalized to the neonatal time point for each cluster, reflecting the dataset quality with overall CV <10% (*Figure 4—figure supplement 1a*). For each cluster, we compute and plot the mean protein abundance (*Figure 4b*). Clusters 1, 2, 4, and 5 show increase protein expression over time, while clusters 3 and 8 are characterized by decreased protein expression over time. Clusters 6 and 7 fluctuate around the baseline with negative and positive peaks during the early postnatal time point, respectively. To track the overall maturation of presynaptic proteins, we mapped our data to SynGO presynapse annotation. For visualization, we plotted heatmaps divided into seven categories: presynapse, cytosol/cytoskeleton/endosome, presynaptic membrane, active zone, endocytic zone, ribosome and ER, and neuronal dense core vesicles (*Figure 4c*, *Figure 4—figure supplement 1b*, *Supplementary file 7*). Majority of presynaptic proteins shows increased expression over development, while presynaptic ribosomal and ER proteins decreased. For quality control, we plotted several examples of known developmentally regulated proteins (*Gonzalez-Lozano et al., 2016*), including doublecortin (DCX), neurofilament medium polypeptide (NEFM), synaptophysin (SYP), Vesicular glutamate transporter 1 (VGLUT1), and vesicle marker Ras-related protein Rab-3A (RAB3A) (*Figure 4—figure supplement 1c*). As expected, an immature neuronal marker like microtubule-associated protein DCX rapidly decreases during development, while mature neuronal markers such as medium intermediate filament (NEFM) and other synaptic and vesicle markers (SYP, VGLUT1, and RAB3A) increase over time. In addition, we performed a meta-analysis comparing our findings with previously reported developmental proteomics studies. We selected three studies for the comparison: cortical synaptic membranes (*Gonzalez-Lozano et al., 2016*) for P70–P9 vs. P50–P5 in this study (adult), forebrain growth cone membrane and particulates (*Chauhan et al., 2020*) for P9–P3 vs. P12–P5 in this study (earlypostnatal), and striatal synaptosomes (*Peixoto et al., 2019*) for P18–P8 vs. P18–P5 in this study (preweanling). We examined the trend of log2FC (e.g. positive or negative) for the overlapping proteome among datasets. We found that >65% of the overlapping proteome between any pair of studies change in the same direction (85% for adult, 65% for earlypostnatal, and 69% for preweanling comparison, respectively) (*Supplementary file 7*). Given extensive methodological differences, sample types, and variance in age across these studies, we consider the results to be a high level of agreement, confirming protein abundance trajectories over development across systems.

Abnormal neurodevelopmental processes including axonal development have been implicated in many psychiatric diseases, with numerous genetic variations and mutations associated with specific disorders (*Werling et al., 2020*). While transcript levels in brain tissues from multiple species have been well cataloged across development (*Colantuoni et al., 2011*; *Werling et al., 2020*), the temporal patterns of protein and protein phosphorylation levels in neuronal subtypes are not fully mapped (*Carlyle et al., 2017*; *Gonzalez-Lozano et al., 2016*), which limits our understanding of how disease risk genes contribute to disease etiology. This question is especially important for our understanding of neuropsychiatric diseases of polygenic origin, where protein networks are believed to serve as the point of convergence for multiple genetic risk factors. To determine whether protein clusters are enriched for disease risk genes, we compiled a list of risk genes associated with 11 disorders: autism spectrum disorders (ASD), Alzheimer's disease (AD), Parkinson's disease (PD), epilepsy (Ep), developmental delay (DD), Schizophrenia (Scz), multiple sclerosis (MS), major depressive disorder (MDD), bipolar disorders (BP), attention deficit hyperactivity disorder (ADHD), and glioma (*Supplementary file 8 Abrahams et al., 2013*; *Chang et al., 2017*; *Trubetskoy et al., 2022*; *Study, 2017*; *Demontis et al., 2019*; *Fu et al., 2021*; *Heyne et al., 2018*; International Multiple Sclerosis *International Multiple Sclerosis Genetics Consortium, 2019*; *Rice et al., 2016*; *Stahl et al., 2019*; *Wightman et al., 2021*; *Wray et al., 2018*). We used hypergeometric testing implemented in Webgestalt (*Wang et al., 2017*) to statistically evaluate risk gene enrichment using the 2107 maSigPro quantified proteins as the background. Clusters with p-value < 0.5 and FDR < 0.3 were considered significant. We found that clusters 2, 3, 5, and 6 were enriched in ASD, BP, EP, and AD risk genes, respectively (*Figure 4d*, *Supplementary file 9*). In addition, we performed Reactome pathway enrichment analysis (*Jassal et al., 2020*) to look at which cellular pathways associate with each protein cluster (*Figure 4e*, *Supplementary file 9*). We examine the relationship between risk genes and biological pathways in the

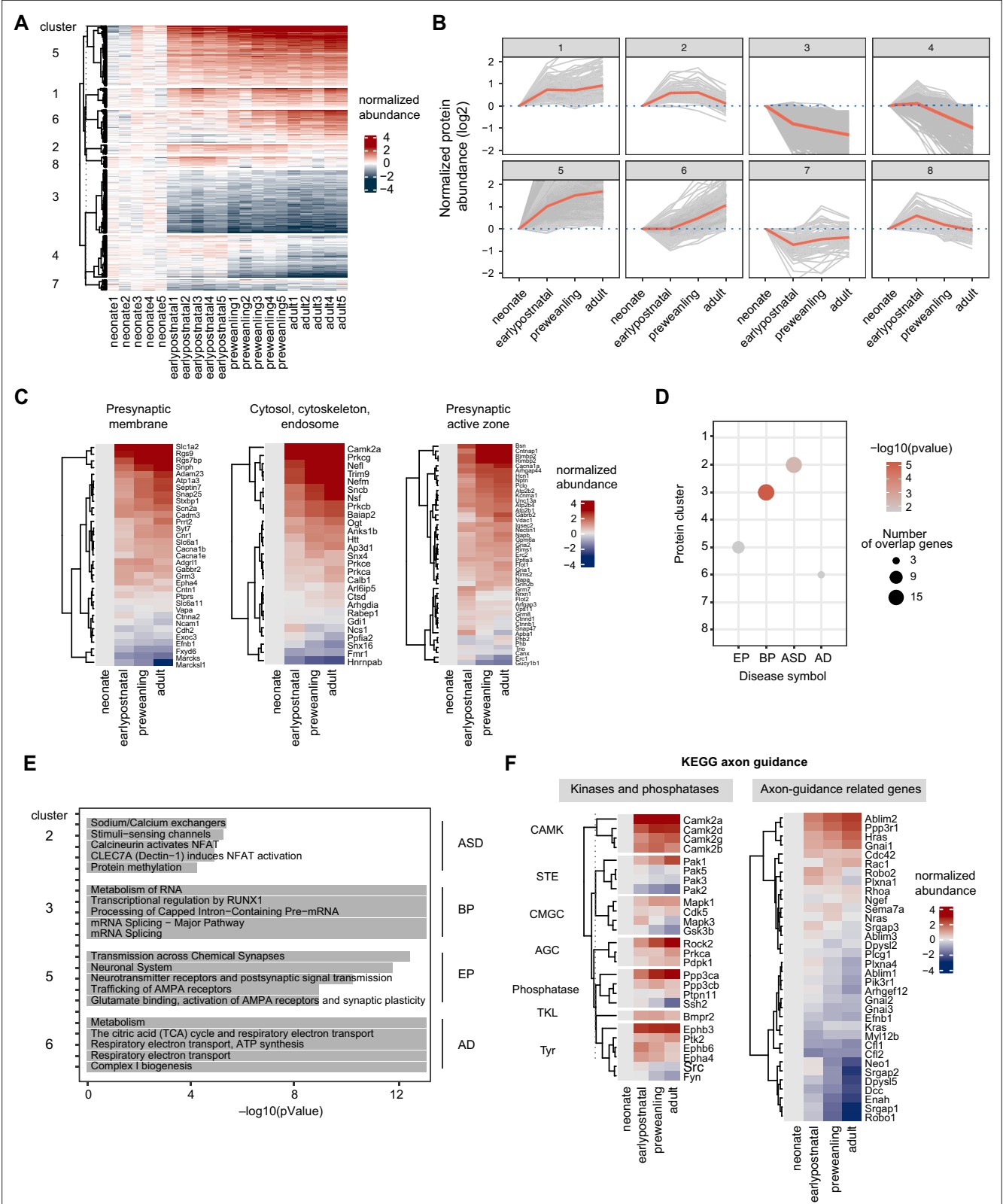

**Figure 4.** Temporal trajectories of the corticostriatal axonal proteome. (**A**) Heat map showing temporal expression across development. Protein clusters were defined using a time course regression analysis implemented in maSigPro R package. The row dendrogram shows hierarchical clustering of proteins in each protein cluster. Log2 protein abundance was normalized to the mean of the neonatal time point. All biological replicates are shown in columns. (**B**) Temporal trajectories of protein expression in each cluster. Log2 protein abundance was normalized to the mean of the neonatal time

*Figure 4 continued on next page*

*Figure 4 continued*

point. Red lines represent the average expression of all proteins in a cluster. Gray lines represent individual protein trajectories. Neonate (P5), early postnatal (P11), preweanling (P18–20), and adult (P50). (**C**) SynGO analysis of presynaptic protein maturation, annotated by gene names. Proteins mapped to SynGO presynapse ontology term were grouped into seven categories: presynapse, cytosol-cytoskeleton-endosome, presynaptic membrane, presynaptic active zone, presynaptic endocytic zone, presynaptic ribosome and ER, and neuronal dense core vesicle. Heatmap shows log2 protein abundance normalized to neonatal time point. Three categories, presynaptic membrane, presynaptic active zone, and presynaptic endocytic zone are shown. Remaining categories are in *Figure 4—figure supplement 2*. (**D**) Over-representation analysis of protein clusters for central nervous system (CNS) traits and disorders. Dot size indicates the number of genes enriched for a particular disease. Color density indicates the degree of significance, –log(pvalue). Only significantly enriched diseases are shown. AD (Alzheimer's disease), EP (Epilepsy), BP (bipolar disorders), and ASD (Autism spectrum disorders). (**E**) Reactome pathway enrichment analysis for protein clusters that are associated with CNS traits and disorders (clusters 2, 4, 7, and 8). Top 5 Reactome pathway terms are plotted. Gray bar indicates –log(p value). (**F**) KEGG axon guidance pathway analysis, annotated by gene names. Heatmap of normalized Log2 protein abundance for proteins mapped to the KEGG axon guidance pathway (mmu04360). *Left*, protein abundance for kinases and phosphatases clustered by family. *Right*, other proteins in the pathway.

The online version of this article includes the following figure supplement(s) for figure 4:

**Figure supplement 1.** Sample coefficients of variation and expression of selected developmental markers.

**Figure supplement 2.** Protein network and Netrin-DCC pathway analysis.

cluster using STRING-DB interactome analysis (*Figure 4—figure supplement 2a*). For example, we found that epilepsy risk genes are significantly enriched in protein cluster 5 (p-value = 0.022), which is composed of proteins with increasing expression over time. This implies that postnatal increase in these gene products is important for proper neural circuit development. Cluster 5 also contains genes enriched in α-amino-3-hydroxy-5-methyl-4-isoxazolepropionic acid receptor (AMPAR) trafficking and plasticity (p=1.1e-9), linking epilepsy risk genes in this cluster to excitatory glutamatergic transmission. In fact, reduced corticostriatal excitatory transmission to fast-spiking interneurons was found to trigger absence seizures in *Stxbp1* haplodeficient mouse model and this effect was mitigated by pharmacological activation of AMPAR in striatum (*Miyamoto et al., 2019*). Altogether, this analysis framework generates an overview of how cellular pathways, developmental trajectories, and genetic risk for neuropsychiatric diseases relate to one another in the context of normal axon development in the striatum.

## Pathway-centric analysis for the axon guidance system

To gain deeper insights into the regulation of axon development, we performed pathway-centric analyses focusing on signaling pathways during axon guidance. We mapped our proximity labeling axon proteome data to the KEGG axon guidance pathway (*Kanehisa et al., 2017*). Because kinases and phosphatases are the major regulators of axonal development, we classified mapped genes into two categories: kinases/phosphatases, and axon-guidance related genes. Temporal trajectories of KEGG genes were visualized in heatmaps with row clustering, separating kinases and phosphatases by their families (*Figure 4f*). We found that the majority of kinases increase in expression over time, such as CAMK and AGC families. However, some kinases (e.g. PAK2, PAK3, PAK5, GSK3B, SSH2, SRC, and FYN) show elevated expression during neonatal and early postnatal time points, but decrease in expression over time, suggesting a functional relevance for early axon guidance. In addition, we detected kinases that are known for their crucial roles in neurodevelopment, including the MAPK family, GSK3B, CDK5, and SRC/FYN. MAPK, GSK3B, and CDK5 are proline-directed kinases while SRC and FYN are non-receptor tyrosine kinases. As for other axon guidance related KEGG genes, we can group them by their functional similarities including ligand-guidance receptor systems, GTPases and their regulators, and cytoskeleton remodeling proteins. In particular, small GTPases including RHOA, RAC1, and CDC42 are important actin cytoskeleton regulators that control axon growth cone expansion during development (*Shekarabi et al., 2005*).

As an example of mining the datasets generated here, we show the Netrin1 (NTN1)-DCC subnetwork in the KEGG axon guidance pathway to investigate whether corticostriatal projections follow a canonical model of axon development (*Figure 4—figure supplement 2b*). Dcc is a netrin receptor that expresses in the axon (*Vosberg et al., 2020*). Activation of the DCC pathway by netrin promotes axon outgrowth by recruiting regulators of actin cytoskeleton (*Shekarabi et al., 2005*). This process depends on FYN-mediated phosphorylation of DCC (*Meriane et al., 2004*) as well as SRC-mediated phosphorylation of PLCG1 (*Kang et al., 2018*). Our data show that DCC expression decreases over

time, reflected by negative log2 fold change in the early postnatal and adult samples compared to the neonate. Similarly, SRC and FYN kinases are upregulated during the early postnatal period and downregulated in the adulthood. The timing of SRC and FYN upregulation in the early postnatal time coincides high DCC expression, implicating SRC and FYN as primary regulators of DCC signaling in the neonatal and early postnatal time windows, but not in the adult with matured striatal circuits. Our data for corticostriatal systems are in an agreement with previous reports examining axonal DCC signaling in *Xenopus laevis* retinal ganglion cells (*Meriane et al., 2004*) and in mouse midbrain dopaminergic neurons (*Kang et al., 2018*), validating these data as an important resource for evaluating axonal proteomes.

## Phosphosite-centric analysis of kinase substrate interactions

Measuring protein levels in time and space offers many clues as to how a biological system matures across development and how it can break down in neuropsychiatric disease. However, it cannot measure protein state (e.g. activity) which is regulated in large part by post translational modifications. Even the abundance of protein kinases alone is insufficient to infer their activity and their constellation of downstream phosphorylation; thus, a direct measurement of protein phosphorylation is essential. This is especially important in the context of axon development, where coordinated phosphorylation events control axon pathfinding, growth cone elongation, and growth cone collapse. Therefore, during the study design, we aimed to create a workflow that enables quantification of both protein and phosphopeptide abundance from the same samples (*Figure 3c*). During sample preparation, on-bead digested peptides were labeled with TMTPro reagents, desalted, and then phospho-enriched. We found that TMTPro multiplexing provides sufficient phosphopeptide signal after enrichment (*Figure 5a*). The unbound fraction can then be used to measure relative protein abundance, as confirmed by the analysis in previous sections (*Figures 3–4*).

Here, we only focused on phosphopeptides that were identified with high confidence (1% FDR), with unambiguous phosphosites, and with quantified proteins. After stringently remove peptides with any missing TMT intensity in any sample, we quantified 2353 unique phosphopeptides that were mapped to 708 proteins (*Supplementary file 10*). Batch effect was corrected with internal reference standard normalization. Global median normalization was used to align sample medians (*Figure 5— figure supplement 1a*). Many proteins have more than one quantified phosphopeptide, with a median of 2 phosphopeptides per protein (*Figure 5—figure supplement 1b*). Using the same maSigPro analysis used for proteomic analysis, we divided phosphopeptides into 8 clusters. Clusters 3, 4, and 7 show overall increase in phosphorylation, while clusters 1, 2, 6, and 8 show overall decrease in phosphorylation over development (*Figure 5b–c*). Next, we used motif analysis to look for phosphopeptide sequences that are enriched in our dataset. Motif-x implementation in R (i.e. rmotifx R package) (*Wagih et al., 2016*) was used to identify over-represented motifs against the PhosphoSitePlus as the background database (*Hornbeck et al., 2015*). We found that the top 3 enriched motifs are sequences that can be phosphorylated by proline-directed kinases (*Figure 5—figure supplement 1c*). This is consistent with our analysis of the axon guidance pathway and previous reports implicating proline-directed kinases such as MAPK family, GSK3B and CDK5, in neurodevelopment (*Igarashi and Okuda, 2019*).

To incorporate phosphorylation data in the context of axon development, we aim to build a signaling model that can infer upstream kinases in the context of the data, while taking into account developmental timepoint to identify functional kinase-substrate interactions. We used PHOsphorylation Networks for Mass Spectrometry (PHONEMeS) analysis tool to model our data (*Gjerga et al., 2021*). PHONEMeS reconstructs phosphorylation signaling networks from known kinase-substrate interaction database (i.e. prior knowledge network, PKN), taking into account 'perturbed' (differentially detected) phosphosites across time points. With respect to a chosen upstream target kinase, PHONEMeS models phosphorylation signaling propagation from the chosen target node to reach the perturbed phosphosites. Our prior analysis of the NTN1-DCC pathway points towards SRC and FYN kinases as key regulators of the axon guidance signaling. To better understand the axon guidance pathway, we focused on FYN as our target kinase that is significantly upregulated in the earlier time windows *Figure 5—figure supplement 1d*. For PHONEMeS input, phosphosites that are differentially detected from neonate were considered as perturbed nodes for that time point (e.g. early postnatal – neonate). PHONEMeS allows users to configure background network to a species other than human.

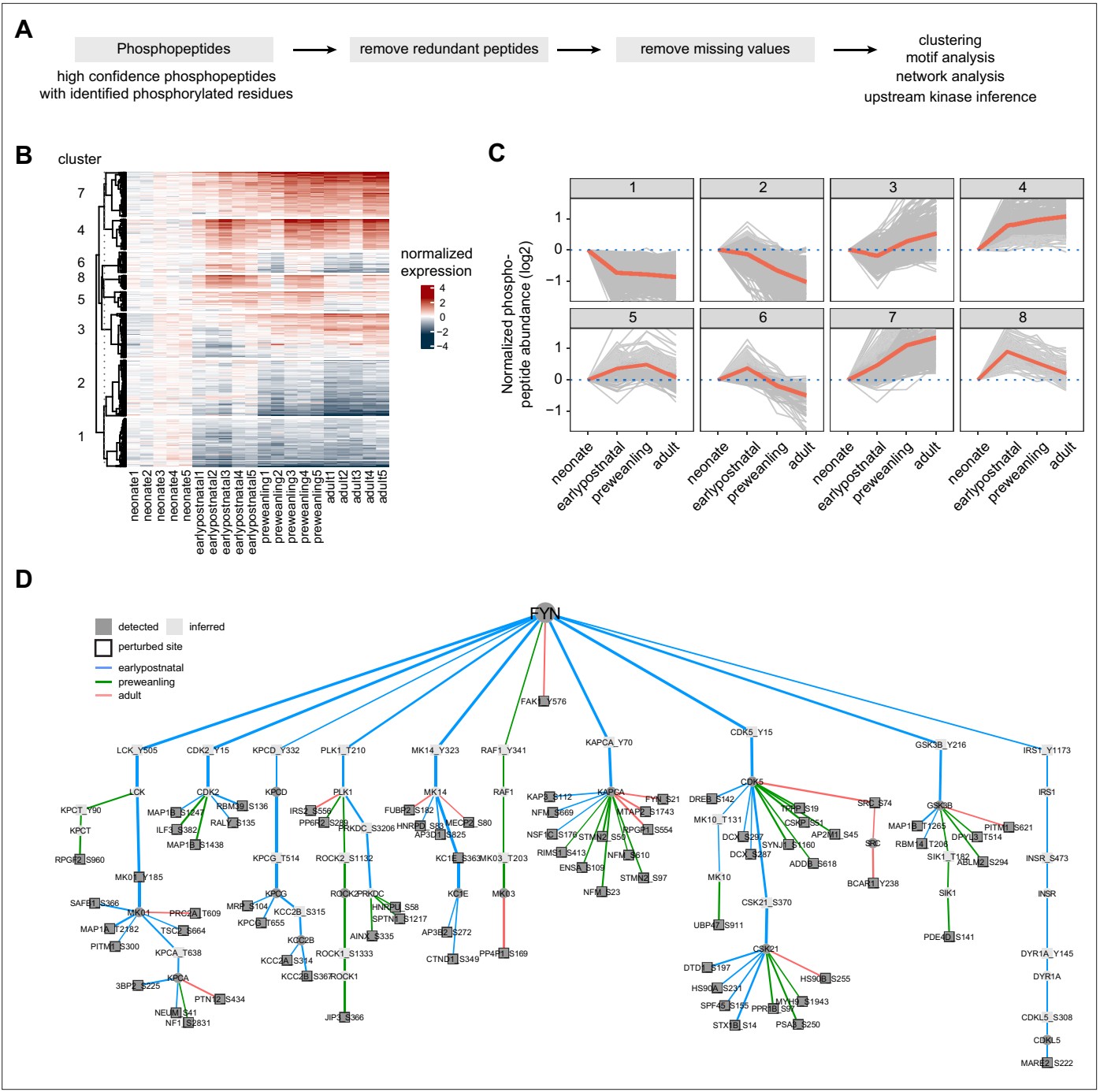

**Figure 5.** Temporal trajectories of phosphopeptide abundance and kinase-substrate interaction network. (**A**) Phosphopeptide analysis workflow. Raw MS data were analyzed by Proteome Discoverer with false discovery rate (FDR) set at 1% for peptide spectrum matches (PSMs). Only phosphopeptide groups with identified phosphorylated residues were used in the analysis. A unique phosphopeptide is defined by a combination of phosphopeptide and its corresponding protein. Entries without missing abundance values across replicates were used in the timecourse and clustering analysis. (**B**) Heat map showing temporal expression of phosphorylation across development. Peptide clusters were defined using a time-course regression analysis implemented in maSigPro R package. The row dendrogram shows hierarchical clustering of peptide in each cluster. Log2 peptide abundance was normalized to the mean of the neonatal time point. All biological replicates are shown in columns. (**C**) Temporal trajectories of phosphorylation in each peptide cluster. Log2 peptide abundance was normalized to the mean of the neonatal time point. Red lines represent the average expression of all proteins in a cluster. Gray lines represent individual protein trajectories. Neonate (P5), early postnatal (P11), preweanling (P18–20), and adult (P5). (**D**) Kinase-substrate interaction network downstream of FYN kinase. PHOsphorylation Networks for Mass Spectrometry (PHONEMeS) was used to model phosphosites using known prior knowledge network from PhosphoSitePlus database. Nodes, kinase or phosphosite. Node color: dark gray, detected,

*Figure 5 continued on next page*

*Figure 5 continued*

light gray, inferred by the model. Edges, kinase-substrate interactions. Edge thickness shows the degree of importance inferred for each kinase-substrate interaction. Edge color: blue, green, red for earlypostnatal–neonate, preweanling–neonate, and adult–neonate comparison, respectively.

The online version of this article includes the following figure supplement(s) for figure 5:

**Figure supplement 1.** Phosphopeptide abundance normalization and motif analysis.

This is important because exact phosphorylation residues are not readily mapped across species. We used OmnipathR to obtain known mouse kinase-substrate interactions (*Türei et al., 2016*). We relied on Cytoscape for visualizing PHONEMeS output (*Figure 5d*). The model assigned edge color and thickness based on their significance for that time point, evidenced by the data. From this model, we observed that FYN is important for phosphorylation events in the early postnatal time point, revealed by a greater proportion of blue edges (early postnatal) compared to green (preweanling) and red (adult). We also found that FYN can directly or indirectly regulate proline-directed kinases such as MAPK1 (MK_01), CDK5, and GSK3B, whose phosphorylation motifs are enriched in our dataset (*Figure 5—figure supplement 1e*). Thus, the result is in agreement with our pathway-centric analysis (*Figure 4f*) and motif analysis (*Figure 5—figure supplement 1c*), supporting the model that FYN is one of the key regulators in the axon guidance pathway in the corticostriatal systems. Altogether, phosphosite-centric analysis by PHONEMeS provides a useful roadmap to extract kinase-substrate interactions that are contextualized to the data, as well as inferences about intermediate kinases and their potential roles in the signaling pathways.

## Correlation analysis of phosphopeptide and protein abundance

Phosphorylation plays a crucial role in protein function, including control over enzymatic activity and protein-protein interactions. As protein expression changes, phosphorylation (as a percentage of total protein) may remain steady or change as levels or activity of kinase and phosphatases fluctuate. Our dataset highlights dramatic reshaping of axonal proteome levels across development, which raises the question of how protein activity is altered on top of changing protein levels across development. To address this question, we computed Spearman rank correlations between phosphopeptides and proteins across developmental time (*Figure 6a*, *Supplementary file 11*). Values associated with correlation coefficients from 2109 peptide-protein pairs were corrected, with 10% FDR as the cutoff for statistical signficance. We found that the majority of phosphopeptide intensities correlate with protein abundance, illustrated by a left-skewed histogram (*Figure 6b*). Out of 309 phosphopeptides with a negative correlation to protein level, 59 decorrelated peptides passed the FDR cutoff (*Figure 6c*). To determine the directionality of changes between protein and peptide abundance across time, we calculated log2 fold change for the corresponding protein between adult and neonate for each peptide. Comparing protein log2 fold change with correlation coefficients, we found 17 decorrelated peptides with negative protein log2 fold change, indicating that phosphorylation increases over time while protein abundance decreases. Analogously, 42 decorrelated peptides with positive protein log2 fold change indicate that phosphorylation decreases over time as protein increases.

Next, we investigate whether decorrelated phosphosites are important for axonal development. To narrow down the list, we chose to examine decorrelated phosphopeptides on proteins that are associated with disease risk genes (*Supplementary file 11*). Among decorrelated peptides, 15 phosphosites are mapped to disease risk genes with 12 out of 15 sites conserved in humans. Particularly, we found TSC2_Ser664 in our PHONEMeS model under the regulation of MAPK1 (*Figure 5d*, *Figure 5—figure supplement 1e*). We found that TSC2 protein moderately increases throughout development, while TSC2_Ser664 rapidly drops off after the neonatal period (*Figure 6d*, corr = –0.570, FDR = 0.009). TSC2 is one of the key regulators of the MTOR pathway (*Ma et al., 2005*), and is implicated in autism spectrum disorder. Activation of TSC1/TSC2 complex decreases MTOR activity by inhibiting RHEB, a positive MTOR regulator. When TSC2 is phosphorylated by MAPK1 at Ser664, TSC1/TSC2 complex is destabilized, allowing RHEB to activate MTOR. This interaction between MAPK1 and TSC2_Ser664 during the earlier time point is emphasized by the PHONEMeS model. To evaluate this regulation more closely, we generated a simplified model of TSC2-RHEB-MTOR signaling and plotted relative protein and phosphopeptide abundance normalized to the neonatal time point (*Figure 6e*). In our dataset, relative phosphorylation of TSC2 at Ser664 in the corticostriatal axons is

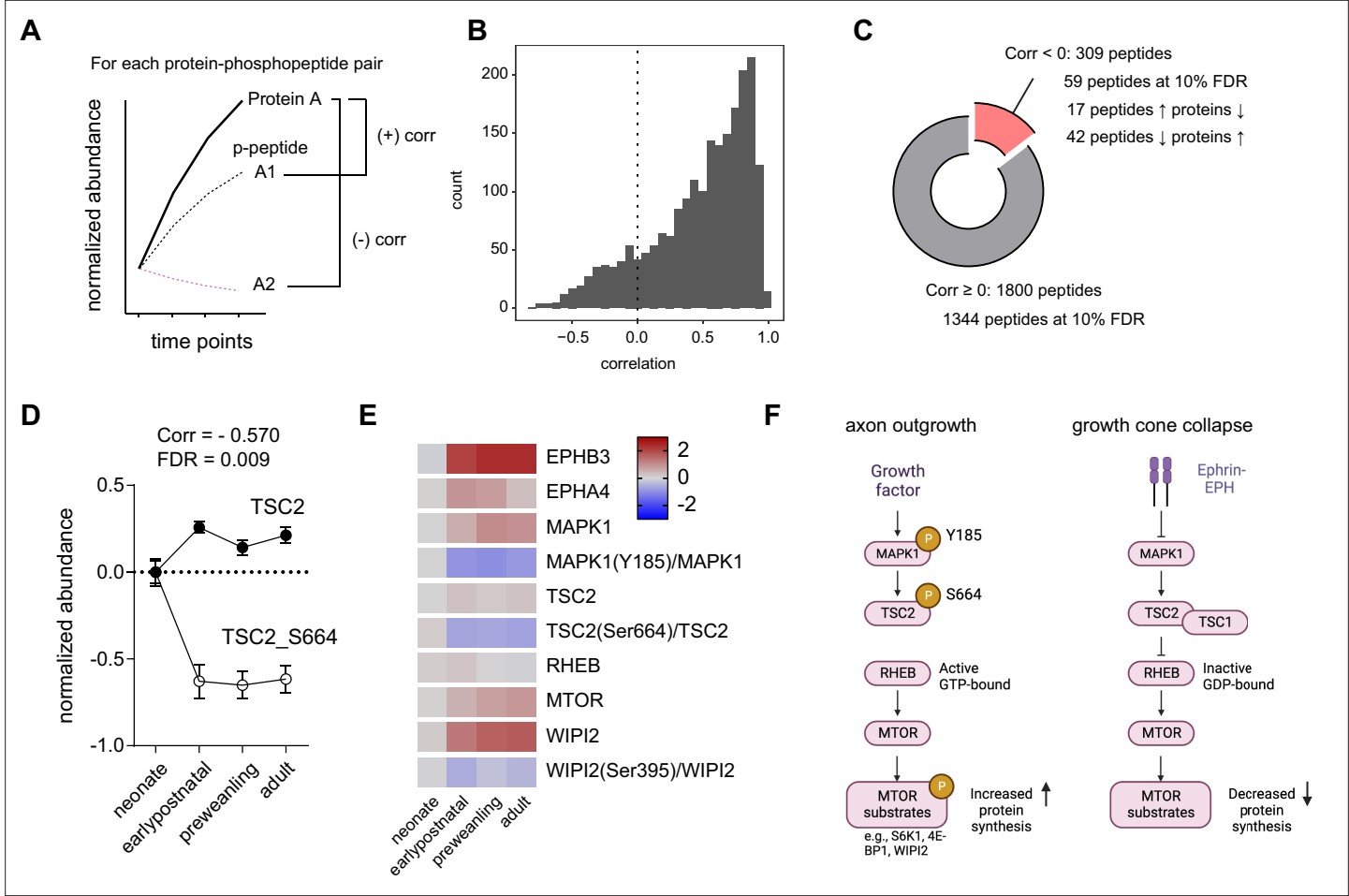

**Figure 6.** Correlation analysis of phosphorylated peptides and proteins implicates Tsc2 Ser664 in MTOR regulation in corticostriatal axon development. (**A**) Correlation analysis workflow. Spearman rank correlation is calculated for each protein and phosphopeptide pair across time points. One protein may have multiple phosphopeptide-protein pairs, yielding multiple correlation values. False-discovery rate was controlled by the Benjamini-Hochberg approach.(**B**) Distribution of protein-phosphopeptide correlations. Left-skewed histogram (bin width = 1). Most phosphopeptides correlate with their respective protein expression across time. (**C**) Classification of protein-phosphopeptide correlations. Cutoff for correlated peptides, Spearman correlation ≥0. Of 1800 phosphopeptides, 1344 are positively correlated with proteins by their temporal expression at 10% false discovery rate (FDR). A small proportion of peptides (59 of 309 phosphopeptides) are decorrelated (inversely correlated) at 10% FDR. 42 phosphopeptides decrease in abundance as protein levels increases, while 17 phosphopeptides show the opposite pattern. (**D**) TSC2 and TSC2_Ser664 expression. Log2 abundance for TSC2 (open circles) and TSC2_Ser664 (solid circles) were normalized to the neonatal time point (Spearman rank correlation: –0.570, FDR = 0.009. n=5 for each time point. Error bars represent SEM).(**E**) Expression heatmap for proteins and phosphosites in the TSC2-RHEB-MTOR signaling pathway. Log2 protein expression was normalized to the neonate time point. Log2 phosphopeptide signal was normalized to the corresponding protein abundance and to the neonate time point. (**F**) Model for MTOR-mediated growth cone collapse.

highest around P5 and decreases over time. Elevated TSC2_Ser664 phosphorylation inhibits TSC1/TSC2 activity, in turn, activating MTOR. We inferred the activity of MTOR by examining its substrate phosphorylation. We detected one MTOR substrate, WIPI2 and WIPI2_Ser395. Consistent with the model, WIPI2_Ser395 phosphorylation is higher in the neonate, supporting the hypothesis that the overall MTOR activity is greater during axon innervation (*Figure 6f*). As neural circuits mature, axonal outgrowth stops and growth cone collapses. A general mechanism for growth cone collapse mediated by ephrin-Eph receptor signaling was studied in the retinal ganglion cells (*Nie et al., 2010*). The activation of EPH receptor leads to inhibition of MAPK1, in turn, allowing TSC2 to inhibit RHEB and MTOR. In the striatum, it has been reported that the growth cones collapse after P7. As expected, we found EPHA4 and EPHB3 Eph receptor expression increase from P5 to P11, coinciding with reduction in TSC2_Ser664 and WIPI2_Ser395 (*Figure 6f*). Therefore, our data support the general mechanism

that MTOR activity is elevated during axon outgrowth and reduced during growth cone collapse, and that this process is conserved in corticostriatal projections.

In summary, we demonstrate a sensitive proximity labeling workflow involving a tandem enrichment of biotinylated proteins and phosphopeptides. We use a newly generated and validated APEX reporter line to map the axonal proteome in the corticostriatal system and establish an analysis framework to examine the subcellular proteome and phosphoproteome across development, generating broadly applicable resources for studying the proteomic landscape of mammalian neural circuits.

## Discussion

Genetically encoded proximity labeling is an effective way to perform cell-type and subcellular compartment specific proteomics in the mouse brain (*Dumrongprechachan et al., 2021*; *Hobson et al., 2022*; *Rayaprolu et al., 2021*; *Sun et al., 2022*). Here, we generated a new Cre-dependent APEX2 reporter line to capture snapshots of the neuroproteome with cell type specificity, illustrated by several genetic crosses including *Vglut2*^Cre, *Vgat*^Cre, and *Rbp4*^Cre. In comparison with biotin ligase-based reporters (e.g. BioID, TurboID), APEX-mediated labeling occurs within 1 hr ex vivo in acute brain sections, while in vivo BioID/TurboID labeling takes several days up to weeks, via continuous biotin supplementation (*Rayaprolu et al., 2021*; *Spence et al., 2019*; *Sun et al., 2022*; *Uezu et al., 2016*). Therefore, the APEX-based approach is more suitable for biological questions that require short temporal windows, while BioID/TurboID can measure in vivo steady-state proteome spanning the duration of biotin administration. Importantly, short $H_2O_2$ treatment during labeling induction may result in undesired oxidative stress signaling, which could be reduced through several additional steps. Transcardial perfusion during acute slice preparation may help remove endogenous catalases and peroxidases in blood vessels. Specialized protective perfusing solution, such as the N-methyl-D-glucamine-based solution (*Hobson et al., 2022*), could provide additional neuroprotection. Transcardial delivery of biotin phenol was found sufficient to induce biotinylation in cardiac tissues (*Liu et al., 2020*), but whether biotin phenol or other APEX substrates can penetrate the blood brain barrier effectively remains to be investigated. These additional steps must be tested for cell classes and tissue types of interest. We speculate that the blood brain barrier could be an obstacle for perfusion-based biotinylation in the central nervous system.

Profiling the axonal proteome or transcriptome is challenging, particularly from complex tissues composed of multiple intermingled cell types. Canonical approaches utilize cell culture models in compartmentalized devices to ensure directional growth (*Cagnetta et al., 2018*; *Chuang et al., 2018*), laser capture microdissection of axon terminals (*Zivraj et al., 2010*), manual dissection of nerve segments (*Michaelevski et al., 2010*), synaptosome sorting from hippocampal mossy fibers (*Apóstolo et al., 2020*), or growth cone (GC) sorting from developing brain (*Chauhan et al., 2020*; *Poulopoulos et al., 2019*). However, no single approach can flexibly measure the overall axonal proteome across a wide age range, with high efficiency, genetic targeting, and pre-synaptic specificity. In this work, we leverage the early expression of APEX in the transgenic reporter cross to map the proteomic landscape of corticostriatal axons from neonate (P5) to young adult (P50). Enabled by temporal precision of APEX-mediated biotinylation, we demonstrate that APEX-based proximity labeling detects dynamically changing proteome of corticostriatal axons across multiple subcellular compartments including axoplasm, synapses, and mitochondria, without the need for biochemical fractionation, or sample pooling beyond P5. It is unexpected that postsynaptic proteins, such as the postsynaptic density 95 protein PSD95, were detected in our axon dataset. One possibility is that such proteins exist in high abundance and are strongly associated with presynaptic partners during streptavidin enrichment, resulting in co-purification. Although it is unlikely, we cannot completely exclude the possibility of leakiness of the APEX2 reporter expression, or the transfer of APEX2 across compartments. Another explanation is that corticostriatal projections form axo-axonic synapses among excitatory inputs (e.g. cortical-cortical or cortico-thalamic axo-axonic synapses), since PSD95 in axons implies glutamatergic presynaptic partners. Indeed, prior ultrastructural observations identified putative glutamate-dopamine axo-axonic contacts in the striatum (*Cover and Mathur, 2021*). Dual color fluorescent tracing also showed rare appositions among thalamic, cortical, and basolateral amygdala inputs onto dendritic spines of spiny projection neurons in nucleus accumbens (*Xia et al., 2020*). Therefore, further ultrastructural studies are needed to determine whether there are glutamaterigic axo-axonic synapses on corticostriatal axons in the dorsal striatum.

Furthermore, we have designed a workflow to add additional steps during the sample preparation for enriching phosphorylated peptides from the same samples, as the ex vivo biotinylation yields sufficient materials. The tandem enrichment workflow enables measurement of both protein and phosphopeptide abundance using TMTPro-based multiplexing and quantification, similar to a recent report (*Zhang et al., 2022*). Although some peptide loss was observed in this strategy, protein quantification in the flowthrough was highly correlated with the unenriched samples. This is crucial for our study, because investigating phosphosite abundance alone without protein abundance can lead to inaccurate interpretation, especially in a system where protein abundance is changing (*Wu et al., 2011*). While our approach profiles phosphosites on biotinylated proteins, Qin et. al., and colleagues recently developed a sequential protein enrichment strategy that combines streptavidin enrichment with functional protein enrichment (e.g. phase separation for RNA-binding proteins, IMAC resin for phosphoproteins) to discover novel functions of a protein subclass in a specific subcellular compartment in HEK cells, highlighting the utility of tandem enrichment in the context of APEX-based proteomics (*Qin et al., 2021*).

Using our workflow, we interrogate axonal proteome of the developing corticostriatal system. We obtained sufficient profiling of the overall landscape. Despite stringent removal of proteins with missing measurements, we were able to perform pathway-centric and phosphosite-centric analyses with good coverage. We showcase an important set of systems-level analyses, encompassing a mapping of protein and phosphopeptide temporal trajectories, disease risk gene association, contextualized kinase-substrate interactions, and correlations between protein and phosphopeptide levels across time, enabling interrogation of complex signaling pathways such as the axon guidance. Molecular level mechanisms of axon guidance systems in the brain have been investigated in a limited number of models. Several examples including Netrin-DCC signaling in dopaminergic (*Kang et al., 2018*; *Vosberg et al., 2020*), thalamocortical (*Castillo-Paterna et al., 2015*) and commissural axon projections (*Varadarajan et al., 2017*), and Ephrin-EPH receptor signaling in retinal ganglion cell projections (*Nie et al., 2010*), callosal axon navigation (*Nishikimi et al., 2011*) and hippocampal mossy fibers (*Xu and Henkemeyer, 2009*). Our pathway-centric analysis reveals that both Netrin-DCC signaling and EPHA signaling are conserved in corticostriatal projections, demonstrating an analysis framework that synthesizes prior knowledge from different neural circuits. Beyond this example, many signaling pathways remains to be explored and our datasets can be used as a general resource to generate testable models for axon development in the context of neurodevelopmental disorders for further mechanistic studies.

Our example of phosphorylation network and correlation analysis emphasizes developmental roles of MAPK1, a proline-directed kinase (PDK), in the MTOR-dependent regulation of axon growth cones. Other developmentally regulated PDKs include CDK5 and GSK3B, well-known regulators of the axon guidance pathway. Deletion of neuron-specific CDK5 activator protein, *p35*, disrupts callosal axons assimilation into the corpus callosum in mice and affects cortical neuron migration and axodendritic orientations (*Chae et al., 1997*; *Kwon et al., 1999*). In contrast, inactivation of GSK3B promotes axon extension (*Zhou et al., 2004*). Deletion of *Gsk3b* in cortical neurons disrupts dendritic orientations, but spares axon elongation into the corpus callosum. Density and arborization of axons in *Gsk3b–/–* neurons remain to be investigated (*Morgan-Smith et al., 2014*). Altogether, proper PDK activity is essential for axonal development.

In this study, we identified FYN as a regulator of PDKs via the kinase-substrate map. *Fyn* knockout (*Fyn* KO) shows cortical, hippocampal, and cerebellar abnormalities in neuronal migration, leading to partial Reeler phenotype and behavior (*Grant et al., 1992*; *Kuo et al., 2005*). In addition, inversion of cortical lamination was observed in *Fyn* KO mice (*Yuasa et al., 2004*). Therefore, we predict that dysregulated cortical stratification in *Fyn* KO mice likely affects cortical projection patterns and striatal function. FYN is also known to regulate CDK5, a PDK that controls actin cytoskeleton remodeling (*Sasaki et al., 2002*). *Fyn* KO resulted in abnormal axodendritic branching of cortical neurons due to decrease in CDK5 tyrosine phosphorylation and activity (*Sasaki et al., 2002*). The same CDK5 phosphosite was found in our kinase-substrate network, highlighting *Fyn* as a regulator of cell migration and subcellular morphology development. However, the specific role of *Fyn* in the development of corticostriatal connectivity is understudied, and to what extent *Fyn* deletion affects other PDK functions in axons remains unclear. Generation of conditional *Fyn* KO in corticostriatally projecting neurons would be informative to elucidate its role in striatal development.

Altogether, our analyses show signaling crosstalk among PDKs and their regulators in developing brain.

Data generated by this work serve as a general reference for overall corticostriatal afferent development at the proteomic and phosphoproteomic levels. Notably, cortical afferents are classified into pyramidal tract (PT) and intertelencephalic tract (IT) projections (*Shepherd, 2013*). PT projections innervate the striatum in the embryonic period, while IT projections arise postnatally, around P3–P4 (*Nisenbaum et al., 1998*; *Sheth et al., 1998*; *Sohur et al., 2014*). Both types of cortical projections to the striatum are topographically organized, forming functionally and spatially distinct striatal regions to process and integrate cortical inputs (*Hooks et al., 2018*). With appropriate Cre-driver lines (e.g. *Tlx3*^Cre_PL56, *Sim1*^Cre_KJ18 for IT and PT neurons, respectively) (*Gerfen et al., 2013*) and timing, analogous approaches can be applied to examine the molecular basis of regional input specific proteome from PT or IT projections during early postnatal development. As a proof of principle in this study, we focus on corticostriatal axons during normal development, but axon maturation requires both presynaptic and postsynaptic interactions. The timely formation of striatal circuits including the maturation of striatal neurons depends on proper levels of synaptogenesis and dendritic spinogenesis, regulated by recurrent network activity, neuromodulation, and experience (*Kozorovitskiy et al., 2012*; *Kozorovitskiy et al., 2015*). Disruption of axon development by disease associated mutations or absence of appropriate dopamine modulation can result in impaired striatal function (*Lieberman et al., 2018*; *Peixoto et al., 2016*). Thus, our work positions the field for genetically targeted proteomics that can be combined with pharmacological, genetic, or engineered effectors to reveal the principles of neural circuit formation between pre- and post-synaptic neurons.

In sum, genetically encoded proteomics tools are rapidly evolving. Together with advances in mass spectrometry, we are starting to obtain sufficient depth of genetically targeted proteome from low microgram samples, with lower rates of missing measurements and with post-translational modification information. With more proteomic datasets of this type, numerous features including cell type, subcellular compartment, developmental trajectory, and phosphorylation can facilitate more accurate theoretical modeling of signaling pathways in the brain during development and in disease contexts. Altogether, methodological and conceptual advancements, along with the datasets generated by this work, provide a new avenue for interrogating the proteomic landscapes of any genetically targeted neural system in the mouse brain.

## Materials and methods
### Mouse strains and genotyping
Animals were handled according to protocols approved by the Northwestern University Animal Care and Use Committee (protocol number: IS00008060). Weanling and young adult male and female mice were used in this study. Cre transgenic lines used in this study include *Vglut2*^Cre (*Slc17a6*^tm2(Cre)Lowl, stock 016963, Jackson Labs), *Vgat*^Cre (*Slc32a1*^tm2(cre)/Lowl, stock 016962, Jackson Labs), and Tg(*Rbp4*^Cre) KL100Gsat/Mmucd (stock 031125-UCD, MMRRC) (*Gerfen et al., 2013*; *Vong et al., 2011*). The Cre-dependent APEX2 mouse line (DIO-APEX2.NES-P2A-EGFP) was generated in this study. The reporter line #83 is submitted to the JAX Repository. C57BL/6 mice used for breeding and backcrossing were acquired from Charles River (Wilmington, MA). All mice were group-housed in a humidity-controlled, ambient temperature facility, with standard feeding, 12 hr light-dark cycle, and enrichment procedures. Littermates were randomly assigned to conditions. All animals were genotyped according to the MMRRC strain-specific primers and protocols using GoTaq Green PCR master mix (cat. no. M712, Promega Corporation, Madison, WI, USA).

## Method details
### Generation of transgenic APEX2 reporter mouse line
DIO-APEX2.NES-P2A-EGFP animals were generated using CRISPR/Cas9 approach at the University of Michigan transgenic core. Mice were housed on ventilated racks or in static microisolator cages with access to food and water with a standard 12 hr light/dark cycle. HDR template was generated by inserting EF1a-DIO-APEX2.NES-P2A-hGH DNA sequence in pROSA26-delDTA between the left and right homology arms. pROSA26-delDTA plasmid was generated by removing the DTA sequence from pROSA26-PA (a gift from Dr. Frank Costantini, Addgene #21271) by cutting with XhoI and SalI,

followed by ligation with T4 DNA ligase. crRNA targeting the *Gt(ROSA)26* locus (*Chu et al., 2016*), 5'-ACTCCAGTCTTTCTAGAAGA-3' was annealed with tracRNA to form duplex gRNA according to manufacturer's instructions (Integrated DNA Technologies). The gRNA was combined with enhanced specificity eSpCas9(1.1) protein resuspended in the provided buffer (cat. no. ESPCAS9PRO, Sigma, MO, USA) to form a ribonucleoprotein complex. The pROSA26-delDTA-EF1a-DIO-APEX2.NES-P2A-EGFP-hGH DNA donor plasmid was purified with for microinjection with an endotoxin-free kit and resuspended in RNAse free microinjection buffer (10 mM Tris, 0.25 mM EDTA, pH 7.4). A test injection was performed with the DNA donor to ensure that it did not interfere with the in vitro development of mouse zygotes from the fertilized egg stage through to the blastocyst stage. The DNA donor plasmid was mixed the CRISPR reagent immediately prior to microinjection into mouse zygotes. The final concentration of the reagents was 7.5 ng/µl crRNA + 5 ng/µl tracrRNA (annealed), 25 ng/µl Cas9 protein, 10 ng/µl circular DNA donor plasmid.

Mouse zygote microinjection was carried out as essentially as described (*Becker and Jerchow, 2011*). Briefly, mixed CRISPR reagents and DNA donor were microinjected into the pronuclei of fertilized eggs at room temperature in hanging drop chambers made of M2 medium (cat. no. M7167, Sigma) surrounded by mineral oil to prevent evaporation. Microinjection needles were pulled from baked (200 °C, 2 hours) 1 mm O.D., 0.75 mm I.D glass capillaries with internal filaments (cat. no. MTW100F-4, World Precision Instruments) on a Sutter P-87 pipet puller. Microinjection needles were connected to a Femtojet pneumatic microinjector (Eppendorf). Zygote holding pipets (200 µm O.D., 30 µm I.D.) were fashioned by hand on a microforge from the same glass capillaries and connected to an Eppendorf Cell-Tram Ari. The micromanipulation workstation consisted of a motorized Nikon TE2000–S microscope (Nikon) equipped with differential interference contrast optics and hydraulic hanging joysticks (Narishige). After microinjection of approximately 3 pl of solution the eggs were washed through four drops of 75 µl $CO_2$ equilibrated KSOMaa medium (Zenith Biotech) under mineral oil. Eggs were surgical transferred to pseudopregnant female mice the same day as they were injected. Fertilized eggs were obtained from the mating of superovulated C57BL/6 J female mice (The Jackson Laboratory stock no. 000664) or B6SJLF1 female mice (The Jackson Laboratory stock no. 100012) with B6SJLF1 male mice. A total of 308 zygotes were microinjected, 273 zygotes were viable after microinjection and transferred to eleven pseudopregnant recipients. A total of 112 pups were born and screened for the *Gt(ROSA)26* insertion.

## Founder screening, genotyping, and transgene characterization

Mouse tail genomic DNA was extracted in 300 µl DirectPCR lysis solution (Viagen Biotech) supplemented with 0.4 mg/ml Proteinase K (Life technologies) at 55 °C overnight. Crude extract was heat inactivated at 85 °C for 45 min and stored at 4 °C. The presence of APEX-transgene was identified using GoTaq Green PCR protocol. Full-length transgene and integration sites were evaluated by Q5 PCR protocol. Detailed PCR methods can be found in *Supplementary file 1*. Potential off-targets of Cas9 activity were identified via Benchling gRNA design tool. The top eight off-targets with *NGG* PAM sequence were selected for analysis. ~800 bp regions flanking the off-target sites were amplified (*Supplementary file 2*). Primers were designed based on sequence retrieved from *Mus musculus* (assembly GRCm38.p6) reference genome on the NCBI database. Twenty µl of PCR product was cleaned up by 2 µl of ExoSAP-IT express reagent for 30 min at 37 °C and 2 min at 80 °C. Sanger sequencing was used to identify mutations (indels or point mutations). All sequence alignment was performed in SnapGene software.

## Stereotactic injections

Conditional expression of H2B.APEX in corticostriatal projection neurons was achieved by recombinant adeno-associated viral neonatal transduction encoding a double-floxed inverted open reading frame (DIO) of target genes, as described previously (*Dumrongprechachan et al., 2021*). For neonatal AAV delivery, P3-6 mice were cryoanesthetized and were placed on a cooling pad. 400 nl of AAV was delivered using an UltraMicroPump (World Precision Instruments, Sarasota, FL) by directing the needle + 1 mm anterior of bregma,±0.15 mm from midline, and 0.8–1.0 mm ventral to skin surface. Following the procedure, pups were warmed on a heating pad and returned to home cages, with approved post procedure monitoring. AAVs were diluted to the final titers using Gibco PBS pH 7.4 (AAV1.EF1a.DIO. H2B-APEX-P2A-EGFP titer ~3 × $10^{12}$ GC/ml). P35-P70 animals were used for histology, western blots,

and proteomics experiments. Cre-negative animals (APEX2 reporter alone) were used as a negative enrichment control group for proteomics, because they do not express APEX transgenes.

## Acute slice preparation and ex vivo biotinylation

Animals were deeply anesthetized with isofluorane inhalation. The brain was removed and placed directly into ice-cold carbogenated (95%$O_2$/5%$CO_2$) artificial cerebrospinal fluid (ACSF; 127 mM NaCl, 25 mM NaHCO$_3$, 1.25 mM NaH$_2$PO$_4$ monobasic monohydrate, 25 mM glucose, 2.5 mM KCl, 1 mM MgCl$_2$, and 2 mM CaCl$_2$). Tissues were blocked and sliced on Leica VT1000S or VT1200S. For ex vivo biotinylation, slices were incubated in carbogenated ACSF with 500 µM biotin phenol (cat. no. LS-3500, Iris Biotech, Germany) at RT for 1 hour. They were briefly rinsed in ACSF and then transferred into ACSF containing 0.03% H$_2$O$_2$ for 2 min. The reaction was quenched with ASCF containing 10 mM sodium ascorbate. Prefrontal cortex or striatum were dissected in >10 ml ice-cold ACSF on sylgard coated petri dish. ACSF was removed and tissues were stored at –80 °C until further processing.

## Immunofluorescence staining

Animals were deeply anesthetized with brief isofluorane inhalation and transcardially perfused with 4% paraformaldehyde (PFA) in 0.1 M phosphate buffer saline (PBS). Brains were extracted and post-fixed in 4% PFA overnight. Brain was rinsed three times with PBS. VT1000S was used to generate 40–60 µm sections in PBS for immunohistochemistry or tissue imaging. APEX expressing cells were identified by GFP immunostaining using chicken anti-GFP antibody (1:2000; cat. no. AB13970, Abcam, Cambridge, UK, RRID:AB_300798) and APEX localization was confirmed by mouse anti-FLAG antibody (1:1000; cat. no. A00187-200, Genscript, NJ, USA, RRID:AB_1720813). Sections were incubated in primary antibody solution overnight at 4 °C (0.5% Triton PBS). Tissues were rinsed three times with PBS (5 min each) and incubated in secondary antibody solution at RT for 1 hr (PBS 1:500 goat anti-chicken Alexa Fluor 488 (RRID:AB_2534096), and goat biotinylated anti-mouse, Thermo Fisher). For detecting FLAG-tagged APEX, tissues were incubated in 1:500-streptavidin Alexa Fluor 647 (Thermo Fisher) at RT for 1 hr. Tissues were then rinsed 3 x (5 min each) with PBS, air dried on Superfrost Plus slides and coverslipped under 10% glycerol-TBS with 2 µg/ml Hoechst 33342 (Thermo Fisher). Sections were then imaged with Olympus VS120 microscope system and/or Leica SP5 confocal microscope.

To verify APEX activity and subcellular localization, tissue sections were incubated in PBS containing 500 µM biotin phenol for 30 min and treated with PBS containing 0.03% H$_2$O$_2$ for 1 min. The reaction was quenched 3 x with PBS containing 10 mM NaN$_3$ and 10 mM sodium ascorbate. Following biotinylation, tissues were immunostained with 1:2000-chicken anti-GFP, 1:1000-rabbit anti-DARPP32 overnight at 4 °C (cat. no. 2306, Cell Signaling, Danvers, MA, USA, RRID:AB_823479). Tissues were rinsed three times with PBS (5 min each) and incubated in secondary antibody solution at RT for 1 hr (PBS 1:500 goat anti-chicken Alexa Fluor 488 (RRID:AB_2534096), goat anti-rabbit Alexa Fluor 594 (RRID:AB_2534079) and streptavidin Alexa Fluor 647, Thermo Fisher).

## Western blot analysis

Total protein was extracted by sonication in 400 µl lysis buffer (1% sodium dodecyl sulfate (SDS), 125 mM triethylammonium bicarbonate (TEAB), 75 mM NaCl, and Halt protease and phosphatase inhibitors). Lysates were cleared by centrifugation at 12,000 g for 15 min at 4 °C. Supernatant was transferred to a new tube and used for subsequent procedure. Total protein (1 µl of samples were diluted to 100 µl with water) was estimated using microBCA assay according to the manufacturer instructions (cat. no. 23235, Thermo Fisher).

Protein lysate inputs or flowthrough fractions were mixed with 6x Laemmli loading buffer and heated to 90–95°C for 10 min. For eluting biotinylated protein off the beads, washed beads were mixed with 20 µl of 2x Laemmli buffer containing 25 mM TEAB, 75 mM NaCl, and 20 mM biotin. Beads were heated to 90–95°C for 10 min. Proteins were separated in 10%, 12%, or 4–20% gradient gels (cat. no. 4561096, Biorad, CA, USA) and transferred to nitrocellulose membrane (cat. no. 926–31090, LI-COR, NE, USA). Blots were briefly rinsed with TBS. For detection of biotinylated proteins, blots were incubated in TBST (0.1% Tween-20) containing streptavidin CW800 (1:10,000, cat. no. 926–32230, LI-COR) for 1 hr at RT. Blots were washed three times with TBST for 10 min each. Total protein was detected using REVERT 700 according to the manufacturer instructions. For other proteins, blots were blocked with 5% milk TBS for 1 hr and probed with primary antibodies prepared in TBST overnight

at 4 °C (1:2000 for tyrosine hydroxylase Millipore cat. no. AB152, Doublecortin, Thermo Fisher cat. no. 481200, synaptophysin-I Synaptic systems cat. no. 101011, Psd95 Synaptic Systems cat. no. 124011, and 1:1000 for DARPP32 Cell Signaling cat. no. 2306) (RRID for Ab are RRID:AB_390204, RRID:AB_2533840, RRID:AB_887824, RRID:AB_1080428, and RRID:AB_823479, respectively). Blots were washed three times with TBST 10 min each at RT and probed with 1:10,000-secondary antibodies (LI-COR, RRID:AB_10956166, RRID:AB_621843, RRID:AB_10956588, RRID:AB_621842). Blots were scanned using a LI-COR Odyssey CLx scanner. All quantification was performed using LI-COR Image Studio version 5.2. For dot blot analysis, 1 µl (0.2 µg) was dotted on nitrocellulose membrane. Membrane was air dried and probed with 1:20000-streptavidin CW800 and total protein stain REVERT 700.

## Mass spectrometry sample preparation

This protocol was modified from our previous publication (*Dumrongprechachan et al., 2021*). Total protein was extracted by sonication in lysis buffer (1% SDS, 125 mM TEAB, 75 mM NaCl, and Halt protease and phosphatase inhibitors). Lysates were cleared by centrifugation at 12,000 g for 15 min at 4 °C. Supernatant was used for subsequent procedure. Total protein was estimated using BCA assay according to the manufacturer instructions (cat. no. 23235, Thermo Fisher).

Brain lysates (300 µg in 250 µl) were reduced with 20 µl 200 mM DTT for 1 hr and alkylated with 60 µl 200 mM IAA in the dark for 45 min at 37 °C with shaking. No pooling was needed except for P5 neonatal striatal samples. For the P5 age group, lysates from two animals of the same sex were pooled to make 300 µg. Streptavidin magnetic beads (150 µl for each sample) (cat. no. 88816, Thermo Fisher) were prewashed with 1 ml no-SDS lysis buffer and incubated with 250 µl of reduced and alkylated lysates for 90 min at RT with shaking. Enriched beads were washed twice with 1 ml no-SDS lysis buffer, 1 ml 1 M KCl, and five times with 1 ml 100 mM TEAB buffer. Washed beads were digested with trypsin solution (~4 µg in 150 µl 100 mM TEAB) overnight at 37 °C (cat. no. 90058, Thermo Fisher). Digested supernatant was collected. Beads were rinsed with 50 µl 100 mM TEAB. Supernatant were combined. Trace amounts of magnetic beads were removed twice by magnetization and tube changes. 10 µl from each sample was saved for Pierce fluoremetric peptide quantification assay. Remaining peptides were frozen and dried in a vacuum concentrator before TMT labeling. To make reference channel samples, a separate set of P18 *Rbp4*Cre+;APEX+ samples were prepared and pooled to make two 900 µg samples (750 µl, reduced with 60 µl 200 mM DTT, and alkylated with 180 µl 200 mM IAA). Pooled lysates were enriched with 400 µl of streptavidin beads, followed by the same bead washing and tryptic digestion (~8 µg trypsin in 300 µl+50 µl rinse). Eluted peptides were pooled and dried in a vacuum concentrator.

Dried peptide samples were reconstituted in 20 µl 100 mM TEAB and sonicated for 15 min at RT. Briefly, one set of 0.5 mg TMTPro 16plex reagents was warmed up to room temperature and dissolved in 20 µl Optima LC/MS-grade acetonitrile (ACN). 6 µl of ~59 mM TMTPro reagents was added to 20 µl reconstituted peptide according to the experimental design in *Supplementary file 3*. For reference channel sample, peptides were reconstituted in 80 µl. A total of 40 µl was labeled with 12 µl TMTPro 134 N. Labeling was performed at RT for 90 min with shaking at 400 rpm. The reaction was quenched by adding 3 µl 5% hydroxylamine/100 mM TEAB at RT for 15 min with shaking (6 µl for reference channel). All samples (for each TMT set, ~29 µl samples x 15 and 24 µl reference) were mixed equally and dried in a vacuum concentrator.

TMT-labeled peptide mixtures were cleaned up by high pH-reverse phase fractionation (cat. no. 84868, Thermo Fisher). Samples were resuspended and sonicated for 10 min in 300 µl buffer A (LC MS water 0.1% TFA). Resin was packed by centrifugation at 5000 g for 2 min, activated twice with 300 µl ACN, and conditioned twice with 300 µl buffer A. Peptides were loaded once by centrifugation at 3000 g for 2 min. Column was first washed with 300 µl water, and 300 µl 2% ACN 98% triethylamine (TEA). Peptides were eluted in 300 µl 25% ACN 75% TEA and dried in a vacuum concentrator prior to phosphopeptide enrichment.

Phosphopeptides were enriched using the AssayMAP Agilent Bravo instrument with Fe(III)-NTA IMAC resin according to the manufacturer's instructions (Agilent, cat. no. G5496-60085). Peptides were resuspended in 210 µl 80% ACN 0.1% TFA, bath sonicated for 10 min, vortexed for 10 min, and quickly centrifuged. The resin was primed with 50% ACN 0.1% TFA and equilibrated with equilibration/wash buffer 80% ACN 0.1% TFA. Samples were loaded into the syringes then dispensed into the Flow Through Collection plate. Syringes were washed with the Cartridge Wash Buffer discarded

into waste then repeated with Syringe Wash Buffer. Elution buffer was aspirated into the syringes and dispensed into the Eluate Collection plate. Phosphopeptides were eluted in 1% ammonium hydroxide solution and acidified to 1% formic acid for MS analysis. The flow through fraction was collected and fractionated by high pH-reverse phase columns as described above. Peptides were eluted by increasing percentage of acetonitrile in 0.1% triethylamine solution according to the manufacturer instructions (5%, 10%, 12.5%, 15%, 17.5%, 20%, 22.5%, and 25% ACN). All fractions were dried in a vacuum concentrator.

## Mass spectrometry data acquisition and raw data processing

TMT labeled peptides (~1 µg) were resuspended in 2% acetonitrile/0.1% formic acid prior to being loaded onto a heated PepMap RSLC C18 2 µm, 100 angstrom, 75 µm x 50 cm column (Thermo Scientific) and eluted over 180 min gradients optimized for each high pH reverse-phase fraction (*Dumrong-prechachan et al., 2021*). Sample eluate was electrosprayed (2000 V) into a Thermo Scientific Orbitrap Eclipse mass spectrometer for analysis. MS1 spectra were acquired at a resolving power of 120,000. MS2 spectra were acquired in the Ion Trap with CID (35%) in centroid mode. Real-time search (max search time = 34 s; max missed cleavages = 1; Xcorr = 1; dCn = 0.1; ppm = 5) was used to select ions for synchronous precursor selection for MS3. MS3 spectra were acquired in the Orbitrap with HCD (60%) with an isolation window = 0.7 m/z and a resolving power of 60,000, and a max injection time of 400ms. Four µl (out of 20) of the TMT labeled phosphopeptide enrichments were loaded onto a heated PepMap RSLC C18 2 µm, 100 angstrom, 75 µm x 50 cm column (Thermo Scientific) and eluted over a 180 min gradient: 1 min 2% B, 5 min 5% B, 160 min 25% B, 180 min 35% B. Sample eluate was electrosprayed (2000 V) into a Thermo Scientific Orbitrap Eclipse mass spectrometer for analysis. MS1 spectra were acquired at a resolving power of 120,000. MS2 spectra were acquired in the Orbitrap with HCD (38%) in centroid mode with an isolation window = 0.4 m/z, a resolving power of 60,000, and a max injection time of 350 ms.

Raw MS files were processed in Proteome Discoverer version 2.4 (Thermo Scientific, Waltham, MA). MS spectra were searched against the *Mus musculus* Uniprot/SwissProt database. SEQUEST search engine was used (enzyme = trypsin, max. missed cleavage = 4, min. peptide length = 6, precursor tolerance = 10 ppm). Static modifications include carbamidomethyl (C,+57.021 Da), and TMT labeling (N-term and K,+304.207 Da for TMTpro16). Dynamic modifications include oxidation (M,+15.995 Da), Phosphorylation (S, T, Y,+79.966 Da, only for phosphopeptide dataset), acetylation (N-term,+42.011 Da), Met-loss (N-term, –131.040 Da), and Met-loss + Acetyl (N-term, –89.030 Da). PSMs were filtered by the Percolator node (max Delta Cn = 0.05, target FDR (strict) = 0.01, and target FDR (relaxed)=0.05). Proteins were identified with a minimum of 1 unique peptide and protein-level combined q values < 0.05. Reporter ion quantification was based on corrected S/N values with the following settings: integration tolerance = 20 ppm, method = most confident centroid, co-isolation threshold = 70, and SPS mass matches = 65. PSMs from Proteome Discoverer were exported for analysis in MSstatsTMT R package (version 1.7.3).

## Validation of the flowthrough fractions for protein abundance analysis

Acute brain slices from *Rbp4*[Cre];APEX were biotinylated ex vivo, lysed, and enriched for biotinylated proteins (n=3 bioreplicates, 1 mouse per replicate). Two mg of proteins were used in the enrichment with 500 µl streptavidin beads. Beads were digested, and peptides were dried in a vacuum concentration prior to TMTPro0 labeling. The use of TMT labeling here is important to mimic the original experiment because TMT-labeled peptides are more hydrophobic than unlabeled peptides. Samples were resuspended in 50 µl of 100 mM TEAB bath sonicated for 10 min, vortexed at 1400 rpm for 10 min, then quick spun down. Each sample was labeled with 1 mg of TMTpro Zero (100 µl at 10 µg/µl) (cat. no. A44518, Lot. VH309856, Thermo Fisher) at RT for 1 hr and quenched with 5% hydroxyl-amine (8.56 µl added). Samples were dried down and resuspended in 0.1% TFA (bath sonicated for 10 min, and vortexed 1400 rpm for 10 min) for desalting on Pierce desalting spin columns following manufacturer protocol. After desalted the samples were split: 50% for unenriched peptide analysis (UE) and 50% for phosphopeptide enrichment on an AssayMAP Bravo (Agilent) with Fe(III)-NTA IMAC resin. Flowthrough fractions (FT) were collected and dried. UE and FT peptides were prepared in a final volume of 40 µl for label-free MS analysis. Raw MS files were processed in Proteome Discoverer

version 2.4 with the same setting except for label-free quantification. PD peptide group intensity and protein abundance were used for analysis without normalization.

## Evaluation of protein and phosphopeptide abundance and axonal enrichment

Protein summarization and data analysis workflow was adapted from our previous work (*Dumrong-prechachan et al., 2021*). PSMs were exported from Proteome Discoverer and converted into MSstatsTMT-compatible format using PDtoMSstatsTMTFormat function. PSMs were filtered with a co-isolation threshold = 70 and peptide percolator q value < 0.01. For protein quantifications, only unique peptides were used. In addition, only proteins with a minimum of two unique PSMs were quantified (e.g. proteins with 1 unique peptide must have at least two PSMs of different charges to be considered for summarization in the MSstatsTMT). Therefore, not all identified proteins were quantified. Protein summarization was performed in MSstatsTMT with the following arguments: method = LogSum, reference normalization = TRUE, imputation = FALSE.

To remove non-specifically enriched proteins, global median normalization was not performed to reflect enrichment differences between APEX + and APEX-negative control. Instead, we used the moderated t-test implemented in the MSstatsTMT package to make the following comparisons (CTX_H2B – CTX_negative, and STR_P18 – STR_negative). Multiple hypothesis adjustment was performed using the Benjamini Hochberg approach. Proteins were considered positively enriched at 0.5% FDR and 2.5-fold change above Cre-negative control. After removing non-specific enrichment, the extent of axon-enrichment was assessed by moderated t-test comparison between CTX_H2B and STR_P18 samples. For this comparison, protein abundance was normalized at the peptide-level by setting global median normalization = TRUE in the MSstatsTMT function. Batch-effect was corrected using LIMMA removeBatchEffect function, followed by protein-level median normalization. Proteins were considered as axon-enriched at 5% FDR and log2FC (STR_P18 – CTX_H2B)>0. For gene ontology analysis, DAVID v6.8 (https://david.ncifcrf.gov/) was used with identifier = `UNIPROT_ACCESSION` (*Huang et al., 2009*). All quantified proteins were used as background. Axon-enriched and somatic-nuclear protein lists were used as input, and –log10(FDR) was plotted for top GO terms. For rank-based analysis in *Figure 3—figure supplement 2c*, we first obtained the total number of proteins in the dataset, that has been previously annotated with GO-terms related to axons and presynapses, denoted by A. The number of remaining proteins not in A is denoted by B. We calculate the GO annotation rate defined as the cumulative proportions of A and B as a function of rank. The difference in annotation rate (A – B) for each rank reflects the degree of axon-protein enrichment for that rank.

For phosphopeptide analysis, only high confidence phosphopeptides in striatal samples were used. Reference channel normalization and LIMMA batch effect correction were performed as described above. Global median normalization was used to align medians across TMT channels. We considered a phosphopeptide unique if there was only one peptide-protein entry in the dataset. For duplicate entries (e.g. two peptides with the same modified phosphosite, but different additional modifications such as methionine oxidation), an entry with the maximum signal summed across channels was used.

## Time course and clustering analysis with maSigPro

To statistically evaluate whether proteins or phosphopeptides change across postnatal development, maSigPro package (version 1.64.0) was used (*Conesa et al., 2006*). Only quantified protein with complete cases in the striatal samples were used. For each protein, maSigPro uses the least-square linear regression to determine whether protein or phosphopeptide abundance significantly changes across four time points. P-value of the F-statistics for each ANOVA (i.e. proteins or unique phosphopeptides) were corrected with Benjamini Hochberg method. Protein or phosphopeptides of similar expression across time were grouped by maSigPro clustering approach (n=8 clusters) using the regression coefficients and hclust method, providing clustering analysis that is based on time course, not abundance alone. Correlation analysis was performed for each phosphopeptide and the corresponding parent protein. Spearman rank correlation was calculated across developmental time. Values

associated with correlation were corrected by Benjamini Hochberg method. The R script can be found in the Kozorovitskiy lab github account (https://github.com/KozorovitskiyLaboratory/STRaxon, copy archived at swh:1:rev:081cca42d806d4175e4a85aea11f34a2642114ba, *Dumrongprechachan, 2022*).

For SynGO analysis of presynaptic proteins, presynapse (GO:0098793) ontology term data were downloaded. Entries were re-classified into 8 categories so that one gene is belonged to one category where child-terms were prioritized. In our dataset, 218 proteins were mapped to presynapse SynGO database (*Supplementary file 7*).

## Overrepresentation analysis (ORA) for genes associated with CNS disorders and Reactome pathway

Hypergeometric test implemented in Webgestalt (http://www.webgestalt.org/) was used to test whether protein clusters were overrepresented by any CNS disorders. Lists of risk genes for Autism, Developmental Delay, Epilepsy, ADHD, Schizophrenia, Major Depressive Disorder, Multiple Sclerosis, Parkinson's Disease, Alzheimer's disease, and Glioma (*Supplementary file 8*) were compiled from recent reports of large GWAS studies of each disorder. Uniprot accessions from protein clusters were used as input. Quantified proteins with complete cases (no missing values) were used as background. Reactome pathway enrichment was performed in Webgestalt with the same input and background as described above. Other parameters were default parameters.

## Network analysis for STRING-DB, KEGG axon guidance and phosphosite analysis

For the KEGG pathway analysis, the axon guidance pathway was downloaded from the KEGG website (https://www.genome.jp/pathway/mmu04360). Protein abundance was normalized to the neonate condition. We mapped KEGG genes back to Swissprot and retained entries that were detected in the dataset. We plotted the data as two heatmaps: kinases/phosphatases, and other axon-guidance related proteins. Kinases and phosphatases were identified by cross referencing the protein list with the Uniprot protein kinase family database (https://www.uniprot.org/docs/pkinfam) and the human dephosphorylation database (DEPOD, http://www.depod.bioss.uni-freiburg.de/) (*Duan et al., 2015*). Interactions related to the Netrin1-Dcc subnetwork and log2FC were entered manually into Cytoscape for visualization. For STRING-DB analysis in *Figure 4—figure supplement 1*, to determine if there is any known protein-protein interactions between the products of autism risk genes and the calmodulin pathway, protein products of the following genes detected in our dataset—*Ank2*, *Ppp1r9b*, *Adcy1*, *Calm2*, *Prkar1b*, and *Prkcg*—were used as the input for the STRING-DB (*Szklarczyk et al., 2021*). Only nodes with connected edges were plotted.

For motif analysis, we used the rmotifx R package (https://github.com/omarwagih/rmotifx, *Wagih, 2022*). PhosphoSitePlus (downloaded in Sept 2021) as the background database and the entire phosphopeptides in our dataset (reformatted into 15-amino acid sequences) as the foreground input. Default parameters were used with serine as the central residue, min.seqs= 20, p-value cutoff = $1e^{-6}$.

For PHONEMES-ILP analysis, we adapted the code published by the Saez-Rodriguez lab github (https://github.com/saezlab/PHONEMeS-ILP, *Gabor, 2022*, *Gjerga et al., 2021*). We used OmnipathR to download the mouse kinase-substrate interactions and filtered for entries that map to Swissprot. This list was used as the prior knowledge network (PKN). Next, we normalized each phosphopeptide to the neonate condition using LIMMA, resulting in the following comparison groups with log2 fold changes and adjusted p-values (earlypostnatal – neonate, preweanling – neonate, and adult – neonate). We defined phosphosites as 'perturbed' if the adjusted p-value for the comparison to the neonate was <0.01. This threshold was chosen so that there were sufficient number of interactions detected in our dataset that are also found in the PKN. We used IBM cplex implementation of PHONEMES-ILP to find network solution across three time points, using FYN as the target node for n=50 iterations. Interactions that were presented in more than 25 sub-sampling iterations were visualized by Cytoscape.

## Materials availability

Requests for resources and reagents should be directed to and will be fulfilled by the Lead Contacts Matthew L. MacDonald (macdonaldml@upmc.edu) and Yevgenia Kozorovitskiy (yevgenia.kozorovitskiy@northwestern.edu).

## Acknowledgements

The authors are grateful to Lindsey Butler for mouse colony management, Dr. Thomas Bozza for advice during mouse line generation, Dr. Thom Saundersm, Wanda Filipiak, and Galina Gavrilina for preparation of gene-edited mice in the Transgenic Animal Model Core of the University of Michigan's Biomedical Research Core Facilities, Northwestern Biological Imaging Facility (RRID:SCR_017767) and Dr. Tiffany Schmidt for confocal microscope access, Northwestern High Throughput Analysis laboratory for the microplate reader, and Dr. Anastasia Yocum and her team at A2IDEA, LLC as data analysis consultants. Some schematics were created with https://biorender.com/. This work was supported by the NSF CAREER Award 1846234, NIMH R56MH113923, NINDS R01NS107539, NIMH R01MH117111, the Beckman Young Investigator Award, Searle Scholar Award, Rita Allen Foundation Scholar Award, and Sloan Research Fellowship (all YK), and NIMH R01MH118497 (MLM). VD is a predoctoral fellow of the American Heart Association (19PRE34380056) and an affiliate fellow of the NIH 2T32GM15538.

## Additional information

### Funding

| Funder | Grant reference number | Author |
| --- | --- | --- |
| National Institute of Mental Health | R56MH113923 | Yevgenia Kozorovitskiy |
| National Institute of Neurological Disorders and Stroke | R01NS107539 | Yevgenia Kozorovitskiy |
| National Institute of Mental Health | R01MH117111 | Yevgenia Kozorovitskiy |
| National Science Foundation | 1846234 | Yevgenia Kozorovitskiy |
| Arnold and Mabel Beckman Foundation | Beckman Young Investigator Award | Yevgenia Kozorovitskiy |
| Kinship Foundation | Searle Scholar Award | Yevgenia Kozorovitskiy |
| Rita Allen Foundation | Rita Allen Foundation Scholar Award | Yevgenia Kozorovitskiy |
| Alfred P. Sloan Foundation | Sloan Research Fellowship | Yevgenia Kozorovitskiy |
| National Institute of Mental Health | R01MH118497 | Matthew L MacDonald |
| American Heart Association | 19PRE34380056 | Vasin Dumrongprechachan |
| National Institute of General Medical Sciences | 2T32GM15538 | Vasin Dumrongprechachan |

The funders had no role in study design, data collection and interpretation, or the decision to submit the work for publication.

### Author contributions

Vasin Dumrongprechachan, Conceptualization, Resources, Data curation, Software, Formal analysis, Funding acquisition, Validation, Investigation, Visualization, Methodology, Writing – original draft, Writing – review and editing; Ryan B Salisbury, Formal analysis, Investigation, Methodology, Writing – review and editing; Lindsey Butler, Resources, Investigation, Methodology, Writing – review and

editing; Matthew L MacDonald, Resources, Data curation, Software, Formal analysis, Supervision, Funding acquisition, Validation, Investigation, Visualization, Methodology, Writing – original draft, Writing – review and editing; Yevgenia Kozorovitskiy, Conceptualization, Resources, Data curation, Formal analysis, Supervision, Funding acquisition, Validation, Investigation, Visualization, Methodology, Writing – original draft, Project administration, Writing – review and editing

## Author ORCIDs
Vasin Dumrongprechachan ⓘ http://orcid.org/0000-0001-5890-6778
Matthew L MacDonald ⓘ http://orcid.org/0000-0002-2222-2996
Yevgenia Kozorovitskiy ⓘ http://orcid.org/0000-0002-3710-1484

## Ethics
Animals were handled according to protocols approved by the Northwestern University Animal Care and Use Committee. (protocol number: IS00008060).

## Decision letter and Author response
Decision letter https://doi.org/10.7554/eLife.78847.sa1
Author response https://doi.org/10.7554/eLife.78847.sa2

---

## Additional files

### Supplementary files
- Supplementary file 1. Primers and PCR protocol for APEX reporter line genotyping.
- Supplementary file 2. Primers and PCR protocol for off-target analysis.
- Supplementary file 3. Sample information and method validation for flowthrough vs. unenriched samples for protein abundance measurement.
- Supplementary file 4. Protein identification and quantification.
- Supplementary file 5. Statistical analysis for axon vs. soma comparison.
- Supplementary file 6. Gene ontology analysis results for axon vs. soma comparison.
- Supplementary file 7. Protein temporal trajectory analysis and SynGO gene ontology.
- Supplementary file 8. List of risk genes for neurodevelopment, neurodegenerative, and neuropsychiatric disorders.
- Supplementary file 9. Disease and Reactome pathway enrichment analysis for different protein clusters.
- Supplementary file 10. Phosphoproteome temporal trajectory analysis.
- Supplementary file 11. Phosphopeptide vs. protein correlation analysis.
- MDAR checklist

### Data availability
Mass spectrometry raw data have been deposited in the PRIDE database (accession number: PXD030864). Code is available on Github (https://github.com/KozorovitskiyLaboratory/STRaxon, copy archived at swh:1:rev:081cca42d806d4175e4a85aea11f34a2642114ba). All analyzed proteomics results are also included as supplementary files. All uncropped gels and blots are included as source data.

The following dataset was generated:

| Author(s) | Year | Dataset title | Dataset URL | Database and Identifier |
|---|---|---|---|---|
| Kozorovitskiy Y | 2022 | Dynamic proteomic and phosphoproteomic atlas of corticostriatal axons in neurodevelopment | https://www.ebi.ac.uk/pride/archive/projects/PXD030864 | PRIDE, PXD030864 |

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
