## [Editor Report]

Knowledge of the protein composition of defined subcellular compartments is of key importance for the characterization of protein machines that mediate defined cellular functionalities. The current paper presents a novel mouse line that will serve as a tool of fundamental value in this context – a Cre-inducible APEX2 reporter mouse line for acute ex-vivo proximity biotinylation. The authors provide compelling evidence documenting the usefulness of the novel reporter line, describing circuit-specific proteomes and phosphoproteomes in the corticostriatal system of the mouse brain during development. The biological insights deduced from bioinformatic analyses of the proteomic data are convincing. The new APEX2 reporter mouse line will be of substantial interest to researchers in many fields of mammalian biology.

---

## [Decision Letter]

**Decision letter after peer review:**

Thank you for submitting your article "Dynamic proteomic and phosphoproteomic atlas of corticostriatal axon neurodevelopment" for consideration by *eLife*. Your article has been reviewed by 3 peer reviewers, one of whom is a member of our Board of Reviewing Editors, and the evaluation has been overseen by Gary Westbrook as the Senior Editor. The following individual involved in review of your submission has agreed to reveal their identity: Alexandros Poulopoulos (Reviewer #2).

The reviewers have discussed their reviews with one another, and the Reviewing Editor has drafted this letter to help you prepare a revised submission.

Essential revisions:

All three reviewers regard this manuscript to be a strong *eLife* candidate. The reviewers and editors propose to transfer the article from the 'Regular Article' to the 'Tools and Resources' section. The main reason is that all reviewers found the 'new biology' provided by the paper to be limited. Further, and as outlined below, the reviewers see a series of critical issues that need to be addressed by the authors before the manuscript could be accepted.

Essential Changes – Possibly Requiring Additional Experiments

1. For sample preparation for mass spectrometry, the authors follow the interesting concept of first enriching the phosphopeptides from the pool of TMT-labeled tryptic peptides and then using the unbound fraction from that step for further peptide fractionation, followed by mass spectrometric protein quantification. While this strategy looks very straightforward in principle, one would expect that the phosphopeptide enrichment comes with an unspecific loss of other peptides in general, and with a semi-specific loss of acidic peptides in particular. Was this potential issue investigated by comparison with samples that were fractionated directly without prior phosphopeptide enrichment? Or with other words: the rationale for this sequential procedure is compelling – quantification of both protein and phosphopeptide abundance from the same (limited) sample, but what is the price for it as to peptide loss? A control experiment regarding the serial phospho-enrichment and peptide fractionation procedure might be helpful here. The reviewers thought of a sample that is biotin-enriched, digested, and TMT labeled (see workflow in Figure 3C). To keep it simple, this can come from pooled brain slices so that limited material is not an issue and it does not necessarily need to have a complex multiplexing design. This sample is split into two: one half is directly going to high pH reverse-phase peptide fractionation, followed by LC-MS; the other half is going through serial phospho-enrichment and peptide fractionation and LC-MS according to the workflow presented. The idea would be to compare the two proteomes obtained, i.e. original input material (half 1) vs. unbound fraction (half 2), eventually by combining the results from the unbound and bound fraction of the phospho-enrichment. This is a suggestion to further corroborate the applicability of the serial procedure, which may be of interest to the phosphoproteomics field in general. However, if the authors can cite cases showing applicability or have similar own data already available from the setup phase of their workflow, the reviewers will not insist on additional experiments as outlined above, as long as the issue is properly addressed.

Essential Changes – Requiring Further Data Analysis

2. The proteomic and phosphoproteomic analysis of corticostriatal axons is interesting but somewhat incomplete. To validate that the approach reliably reports key developmental stages of corticostriatal axons, one should also look at presynapse maturation and map at what time point presynaptic adhesion proteins, vesicle proteins, active zone proteins, and components of the endocytosis machinery appear. The SynGO database might help in such an analysis. What would generally be helpful would be a more systematic presentation of expected proteins in different developmental phases (axon outgrowth, steering*guidance, synaptogenesis, synapse function) vs. the actual findings.

3. Although both male and female animals were used in the study, the authors do not discuss sex differences in the proteome or phosphoproteome during development. If such data are available, seeing differences in proteomics and phosphoproteomics between sexes during development would be very interesting. If there are no observed differences, adding a sentence to clarify this issue would be helpful.

Essential Changes – Material Availability

4. The novel mouse line described here will become a sought-after tool for many fields of mammalian biology. To make this work, the authors must formally state that the line will be made available to all members of the academic research community upon reasonable request, and describe how exactly to obtain it.

Essential Changes – Changes to Text

5. The APEX2 reporter mouse line is a novel tool with broad applicability for proximity labeling approaches and, understandably, the authors advertise its advantages, mainly via the suitability for short temporal windows. However, the discussion on the limitations of the approach falls short. The authors should make clear that the APEX method in general is limited to ex vivo approaches such as the acute brain slices used here due to the limitation that potentially toxic reagents (i.e. low membrane-permeable biotin-phenol and H2O2) have to be delivered to the target tissue. Although treatment with H2O2 is rather short, undesired oxidative stress signaling may have to be taken into account, particularly when protein phosphorylation rather than protein abundance is assessed. It would also be important to discuss the pros and cons of perfusing the mice prior to preparation of brain slices; e.g., in the context of removal of catalases/endogenous peroxidases or potential for substrate delivery (like recently shown in heart, doi 10.1038/s41586-020-1947-z).

6. In line 122, authors may want to change the term "knock-in line" to "reporter line", as Rosa26 was used as a transgene landing-pad locus, rather than for any endogenous regulatory properties of a specific locus, as knock-in may imply to some readers.

7. The authors claim that they "modified published APEX-mediated biotinylation protocols to optimize protein labeling in thick brain tissue (Dumrongprechachan et al., 2021)" (line 150-151), which implies considerable technical advancements in comparison to their earlier work. At first glance, however, it seems that only the incubation time with H2O2 was doubled. The author should expand on their claim – or specify or remove it.

8. As it is essential for understanding the design of the proximity labeling approach, particularly for the non-expert reader, the Rbp4Cre mouse line should be introduced with a few words, instead of just citing the resource paper on GENSAT BAC Cre-recombinase driver lines.

9. Proteins with q-value < 0.05 and log2FC > 0 were considered axonally enriched. Given that the authors handle their data otherwise with highest stringency, is there a special reason not to apply a more stringent/empirically defined threshold on log2FC?

10. When establishing proteins that are enriched in somata vs. axons, a soma prep from the same APEX mouse line would have been more appropriate as a control compared to the virally ovrexpressed Histone 2B-APEX control used, which would label proteins sequestered only to the nucleus. The authors should qualify in the text that proteome differences they see in these two datasets do not arise solely from somatic versus axonal enrichment, but also from the confounding differences of viral versus Rosa26-locus expression levels, and cytosolic versus histone-fused Apex.

11. While there is a series of novelties in this work, overt "first ever" statements in the text (e.g. lines 94-96, 348-349) are redundant and inherently ambiguous in their factuality. The work's novelty is better served speaking for itself.

12. In line 109, the authors should rephrase "revealing proline-directed kinases and phosphosites as major regulators for corticostriatal projection development" to something more closely fitting the observation that proline-directed kinases and phosphosites are dynamically regulated in corticostriatal projections during development. The current phrase infers that the functional component has been determined in this study, which would need further experimental determination (e.g. knockouts). The data provided in the manuscript only point to these proteins being highly dynamic, in both abundance and phosphorylation throughout development.

13. In line 163, authors should change the phrase "differential pattern of protein labeling" to "reduced protein abundance", as there are no obvious ratiometric changes in band intensity patterns in the western blot shown in Figure 2 supplement 1B. If the authors want to make a claim on that, they should provide a GFP blot to normalize to the amount of cells that express APEX in the region to determine if changes in protein abundance are due to the number of VGAT vs VGLUT cells, or due to differential protein abundance.

14. The discussion requires a few additional thoughts: (i) The authors do not properly reflect the involvement of proline-directed kinases in the development of corticostriatal projections, which stands in contrast to the fact that they sell this as one of their major findings throughout the manuscript. (ii) An obvious future research target based on the present study is the role of Fyn in the development of corticostriatal connectivity. Here, a discussion of the Fyn KO phenotype might be informative (as I did not find evidence for changes in corticostriatal connectivity in the Fyn KO). (iii) It is striking that a Gria1-Dlg1-Dlg4 network was discovered in the corticostriatal axon proteomics. This is unexpected – a postsynaptic network in an axonal dataset. The authors should discuss what this might mean for their new mouse line (e.g. leakiness of the APEX2 reporter, transfer of APEX2 etc.). (iv) The discussion on the Netrin1-DCC pathway is not particularly strong, while aspects of the data are stronger. The authors might want to consider either getting rid of this discussion point, or putting Figure 4F in a supplemental figure. The data provided only show phosphorylation events, not activity of a specific pathway, as Fyn is involved in a lot of different processes. Figure 4 supplement 1B may be a more important panel to have in Figure 4.

---

## [Author Response]

Essential revisions:All three reviewers regard this manuscript to be a strong eLife candidate. The reviewers and editors propose to transfer the article from the 'Regular Article' to the 'Tools and Resources' section. The main reason is that all reviewers found the 'new biology' provided by the paper to be limited. Further, and as outlined below, the reviewers see a series of critical issues that need to be addressed by the authors before the manuscript could be accepted.

We thank the reviewers and editors for taking the time to evaluate the manuscript. We agree with the decision to transfer the manuscript from a regular article to the 'Tools and Resources' section.

Essential Changes – Possibly Requiring Additional Experiments1. For sample preparation for mass spectrometry, the authors follow the interesting concept of first enriching the phosphopeptides from the pool of TMT-labeled tryptic peptides and then using the unbound fraction from that step for further peptide fractionation, followed by mass spectrometric protein quantification. While this strategy looks very straightforward in principle, one would expect that the phosphopeptide enrichment comes with an unspecific loss of other peptides in general, and with a semi-specific loss of acidic peptides in particular. Was this potential issue investigated by comparison with samples that were fractionated directly without prior phosphopeptide enrichment? Or with other words: the rationale for this sequential procedure is compelling – quantification of both protein and phosphopeptide abundance from the same (limited) sample, but what is the price for it as to peptide loss? A control experiment regarding the serial phospho-enrichment and peptide fractionation procedure might be helpful here. The reviewers thought of a sample that is biotin-enriched, digested, and TMT labeled (see workflow in Figure 3C). To keep it simple, this can come from pooled brain slices so that limited material is not an issue and it does not necessarily need to have a complex multiplexing design. This sample is split into two: one half is directly going to high pH reverse-phase peptide fractionation, followed by LC-MS; the other half is going through serial phospho-enrichment and peptide fractionation and LC-MS according to the workflow presented. The idea would be to compare the two proteomes obtained, i.e. original input material (half 1) vs. unbound fraction (half 2), eventually by combining the results from the unbound and bound fraction of the phospho-enrichment. This is a suggestion to further corroborate the applicability of the serial procedure, which may be of interest to the phosphoproteomics field in general. However, if the authors can cite cases showing applicability or have similar own data already available from the setup phase of their workflow, the reviewers will not insist on additional experiments as outlined above, as long as the issue is properly addressed.

A recent study by Zhang et al. (Gygi lab) demonstrated a similar sequential enrichment workflow from biotinylated proteins and phosphopeptides (Zhang et al., 2022), although they did not explicitly evaluate this concern for peptide loss. To determine whether phosphopeptide enrichment leads to nonspecific loss of peptides with loss of acidic peptides in particular, we carried out a label-free experiment to directly compare the intensities of TMT-modified peptides before and after phospho-enrichment (New Figure 3 —figure supplement 3a).

Acute brain slices from Rbp4^Cre^;APEX were biotinylated ex vivo, lysed, and enriched for biotinylated proteins. Beads were digested, and peptides were labeled with TMTPro0 and desalted (n = 3 bioreplicates, 1 mouse per replicate). Each sample was then split into two equal volumes: 50% for unenriched (UE) and 50% for phosphoenrichment (PE). UE samples were dried down while PE was phospho-enriched. Flowthrough (FT) fractions from PE were dried down. UE and FT were resuspended in equal volume and analyzed by Orbitrap Eclipse mass spectrometer under the same setting. The use of TMT labeling here is important to mimic the original experiment because TMT-labeled peptides are more hydrophobic than unlabeled peptides. There are two major goals for this experiment: (1) to determine whether the loss of peptides and acidic peptides is significant after phosphopeptide enrichment, and (2) to evaluate whether phosphoenrichment significantly alters protein abundance in the flowthrough fraction compared to the unenriched samples.

First, we examined log2-transformed intensities of TMT-modified peptide groups quantified by Proteome Discoverer. In this analysis, we only considered peptides without any missing values for UE and FT samples (13905 peptide groups). To be conservative, we did not normalize intensity data, relying on sample preparation reproducibility alone (New Figure 3 —figure supplement 3b). Most peptides in this experiment contain at least one acidic residue (D or E) (12060 peptide groups, 86.7%). We used LIMMA to estimate log2 foldchange for each peptide group between UE and FT fractions, accounting for sample pairing (e.g., UE_1 vs. FT_1). Generally, UE and FT are highly correlated at the peptide level (Spearman correlation, R > 0.92) (New Figure 3 —figure supplement 3c). Only 252 out of 13905 peptide groups (1.8%) are significantly changed after phosphoenrichment (123 decrease, and 129 increase in intensities with q-value < 0.05) (New Figure 3 —figure supplement 3d). In addition, there is no apparent relationship between the number of acidic residues and log2 FC (New Figure 3 —figure supplement 3e). Next, we examined whether the distributions of acidic peptides across different bins of log2FC are different from each other (New Figure 3 —figure supplement 3f). We divided log2FC data into 5 equal bins from the lowest log2FC to the highest, plotting histograms of the number of acidic residues per peptide to assess whether the number of acidic residues relates to the change in peptide intensity after phosphoenrichment. Overall, the histograms retain similar distribution across log2FC bins. Thus, overall changes in log2FC observed at the peptide level are not directly related to the number of acidic residues present in the peptide sequence. At this analysis depth, we show that changes in peptide intensities after phosphoenrichment in the flowthrough did not seem to be specific to peptide acidity.

Second, we aimed to determine whether protein abundance quantified in the flowthrough fractions after phosphoenrichment is comparable to that of unenriched samples. We obtained even higher correlation between UE and FT after protein summarization (R > 0.98) (New Figure 3 —figure supplement 3g – h). This suggests that relative protein abundance in FT is comparable to UE. Using LIMMA, we found only 4 out of 2088 proteins (~0.2%) whose abundance was changed after phosphoenrichment (New Figure 3 —figure supplement 3i). A histogram of log2FC for FT–UE is centered at log2FC = 0 (New Figure 3 —figure supplement 3j), validating the use of flowthrough fractions for estimating protein abundance. In addition, the number of PSMs detected per proteins is an important factor protein quantification (New Figure 3 —figure supplement 3k), as expected for any bottom-up proteomics experiments. This could be further improved by high pH reverse phase fractionations and running technical replicates with exclusion list. Altogether, the impact of peptide loss in the flowthrough after phosphoenrichment has limited impact on protein-level summarization in our dataset, supporting the use of this approach.

We have updated the manuscript to reflect these findings. Lines: 220 –235

Essential Changes – Requiring Further Data Analysis2. The proteomic and phosphoproteomic analysis of corticostriatal axons is interesting but somewhat incomplete. To validate that the approach reliably reports key developmental stages of corticostriatal axons, one should also look at presynapse maturation and map at what time point presynaptic adhesion proteins, vesicle proteins, active zone proteins, and components of the endocytosis machinery appear. The SynGO database might help in such an analysis. What would generally be helpful would be a more systematic presentation of expected proteins in different developmental phases (axon outgrowth, steering*guidance, synaptogenesis, synapse function) vs. the actual findings.

We have performed additional analysis to track the maturation of presynaptic proteins by using SynGO ontology terms. We created heatmaps of protein expression by different SynGO presynapse categories to help the reader examine the developmental trajectories of presynaptic proteins found in our dataset (Updated Figure 4c and updated Figure 4 —figure supplement 1b).

We have updated the manuscript to reflect these findings. Lines: 278 – 283

In addition, we performed a meta-analysis comparing our findings with previously reported developmental proteomics studies. We selected 3 studies for the comparison: cortical synaptic membranes (Gonzalez-Lozano et al., 2016) for P70–P9 vs. P50–P5 in this study (adult), forebrain growth cone membrane and particulates (Chauhan et al., 2020) for P9–P3 vs. P12–P5 in this study (earlypostnatal), and striatal synaptosomes (Peixoto et al., 2019) for P18–P8 vs. P18–P5 in this study (preweanling). We examined the trend of log2FC (e.g., positive or negative) for the overlapping proteome among datasets. We found that >65% of the overlapping proteome between any pair of studies change in the same direction (85% for adult, 65% for earlypostnatal, and 69% for preweanling comparison, respectively) (Supplement file 7). Given extensive methodological differences, sample types, and variance in age across these studies, we consider the results to be a high level of agreement, confirming protein abundance trajectories over development across systems.

We have updated the manuscript to reflect these findings. Lines: 290 – 301

3. Although both male and female animals were used in the study, the authors do not discuss sex differences in the proteome or phosphoproteome during development. If such data are available, seeing differences in proteomics and phosphoproteomics between sexes during development would be very interesting. If there are no observed differences, adding a sentence to clarify this issue would be helpful.

Both male and female animals were used in the study. However, the number of replicates for each sex per time point is low (n = 2–3), constraining the value of sex difference analyses. To investigate the overall sex differences across any pair of time points in our dataset, we employed a rank-rank hypergeometric overlap analysis (RRHO). RRHO uses ranked lists of protein log2-foldchange between any two time points for male and female, respectively. The algorithm outputs a heatmap where intensity indicates statistical significance of overlapping protein sets by rank, visualizing trends in gene-expression profiles between any two subcategories (here, male vs. female).

In Author response image 1, we applied RRHO to protein-level data. We observed that most significant overlap between male and female clustered along the diagonal lines. This suggests similar trends in differentially regulated proteins between any two time points. In other words, sets of proteins are regulated in a similar manner (both upregulated in the bottom left quadrant Q3, and both downregulated in the top right quadrant Q1). Therefore, no significant sex difference is observed in our protein dataset.

Analogously, we performed the same analysis for phosphopeptide dataset (Author response image 1). Similar diagonal clusters were observed for most time points, except for between adults and preweanling comparison. Changes in phosphopeptide levels from preweanling to adults showed a greater degree of non-diagonal overlap. Although there was no discernible hotspot in quadrants 2 and 4 (downregulated in female-upregulated in male, and vice versa, respectively), emerging patterns in phosphorylation profiles between male and female across these time points suggest a potential sex difference.

Given the preliminary nature of these sex difference analyses, we consider them appropriate as an Author response image, to be communicated in the publicly available response to review; however, based on Reviewer/Editorial advice, we can integrate this image into figure supplements and add analysis information to manuscript text.

**Author response image 1. sa2fig1:** Rank-rank hypergeometric overlap analysis (RRHO) for sex differences in protein and phosphopeptide levels across development. (A) RRHO analysis for protein abundance across any two time points with male and female as subcategories. Heatmap of –log Pvalue, where higher values indicate greater statistically significant overlapping proteins between male and female. X- and Y axes are normalized rank from highest to lowest log2FC (0 – 1 rank, 0 highest log2FC). Sets of proteins are regulated in a similar manner between male and female if hotspots are in quadrants Q1 and Q3 for both upregulated, or both downregulated, respectively. In contrast, if hotspots are in Q2 and Q4, proteins are differentially regulated between males and females. (B) Same as (A) for phosphopeptides, showing less clustering around the diagonal compared to protein level data.

Essential Changes – Material Availability4. The novel mouse line described here will become a sought-after tool for many fields of mammalian biology. To make this work, the authors must formally state that the line will be made available to all members of the academic research community upon reasonable request, and describe how exactly to obtain it.

We have initiated the process to deposit the mouse line to the Jackson Laboratory that is currently pending review. Members of the academic research community are encouraged to contact Dr. Yevgenia Kozorovitskiy for additional information. Line: 633

“The reporter line #83 is submitted to the JAX Repository.”

Essential Changes – Changes to Text5. The APEX2 reporter mouse line is a novel tool with broad applicability for proximity labeling approaches and, understandably, the authors advertise its advantages, mainly via the suitability for short temporal windows. However, the discussion on the limitations of the approach falls short. The authors should make clear that the APEX method in general is limited to ex vivo approaches such as the acute brain slices used here due to the limitation that potentially toxic reagents (i.e. low membrane-permeable biotin-phenol and H2O2) have to be delivered to the target tissue. Although treatment with H2O2 is rather short, undesired oxidative stress signaling may have to be taken into account, particularly when protein phosphorylation rather than protein abundance is assessed. It would also be important to discuss the pros and cons of perfusing the mice prior to preparation of brain slices; e.g., in the context of removal of catalases/endogenous peroxidases or potential for substrate delivery (like recently shown in heart, doi 10.1038/s41586-020-1947-z).

We thank the reviewer for raising this important discussion point. We have included additional statements about the general limitations of our approach in the Discussion section. Lines: 501 – 509

“Importantly, short H2O2 treatment during labeling induction may result in undesired oxidative stress signaling, which could be reduced through several additional steps. Transcardial perfusion during acute slice preparation may help remove endogenous catalases and peroxidases in blood vessels. Specialized protective perfusing solution, such as the N-methyl-D-glucamine-based solution (Hobson et al., 2022), could provide additional neuroprotection. Transcardial delivery of biotin phenol was found sufficient to induce biotinylation in cardiac tissues (Liu et al., 2020), but whether biotin phenol or other APEX substrates can penetrate the blood brain barrier effectively remains to be investigated. These additional steps must be tested for cell classes and tissue types of interest.”

Here, we provide a preliminary experiment comparing perfusion-based delivery of BP to the ex vivo bath application for our sample type (Author response image 2). We found that perfusion of 15 ml of ACSF supplemented with BP followed by acute slice preparation and H2O2 treatment did not produce detectable biotinylated proteins compared to the control.

**Author response image 2. sa2fig2:** Preliminary test for perfusion-based delivery of biotin phenol (BP). Rbp4^Cre^;APEX+ animals were transcardially perfused with 15 ml ACSF +/- 500 μM BP. Following perfusion, acute slices were prepared and treated with 0.03% H2O2. No difference in streptavidin staining was detected between BP and no-labeling control, while samples that underwent ex vivo biotinylation show robust signal. Total protein loading control was visualized by REVERT stain.

6. In line 122, authors may want to change the term "knock-in line" to "reporter line", as Rosa26 was used as a transgene landing-pad locus, rather than for any endogenous regulatory properties of a specific locus, as knock-in may imply to some readers.

The term “knock-in line” is now changed to “reporter line” in the text. Line: 121

“The reporter line was bred to homozygosity by crossing F2 heterozygotes (Figure 1c).”

7. The authors claim that they "modified published APEX-mediated biotinylation protocols to optimize protein labeling in thick brain tissue (Dumrongprechachan et al., 2021)" (line 150-151), which implies considerable technical advancements in comparison to their earlier work. At first glance, however, it seems that only the incubation time with H2O2 was doubled. The author should expand on their claim – or specify or remove it.

This sentence has been modified. Line: 149 – 150

“We increased BP incubation time to 2 min from previously published APEX-mediated biotinylation protocols to optimize protein labeling in thick brain tissue (Dumrongprechachan et al., 2021), since APEX has been largely characterized and applied in cell culture (Hung et al., 2016; Lobingier et al., 2017; Paek et al., 2017) and small organisms (Chen et al., 2015; Reinke et al.).”

8. As it is essential for understanding the design of the proximity labeling approach, particularly for the non-expert reader, the Rbp4Cre mouse line should be introduced with a few words, instead of just citing the resource paper on GENSAT BAC Cre-recombinase driver lines.

A general description of the Rbp4Cre mouse line has been added in the text. Lines: 183 – 187

“To capture axonal proteome dynamics during this process, we crossed the Rbp4Cre mouse line to our APEX reporter (Gerfen et al., 2013). The Rbp4^Cre^ mouse line expresses Cre recombinase under the control of the retinol binding protein 4 (Rbp4) gene promoter. A previous characterization of this line shows Cre-mediated reporter expression enriched in the accessory olfactory bulb, layer 5 pyramidal tract neurons throughout the neocortex, and hippocampal granule cells in the dentate gyrus (Harris et al., 2014). As expected, Rbp4^Cre^;APEX expression is primarily restricted to layer 5 cortical neurons which also send projections to striatum (Figure 3a).”

9. Proteins with q-value < 0.05 and log2FC > 0 were considered axonally enriched. Given that the authors handle their data otherwise with highest stringency, is there a special reason not to apply a more stringent/empirically defined threshold on log2FC?

With the current threshold, the range of log2 foldchange for axonal enriched proteins is 0.259 – 5.451, implicitly imposing a cutoff at ~1.19 fold change. This threshold was selected based on our prior experience with other subcellularly targeted/enriched datasets.

10. When establishing proteins that are enriched in somata vs. axons, a soma prep from the same APEX mouse line would have been more appropriate as a control compared to the virally ovrexpressed Histone 2B-APEX control used, which would label proteins sequestered only to the nucleus. The authors should qualify in the text that proteome differences they see in these two datasets do not arise solely from somatic versus axonal enrichment, but also from the confounding differences of viral versus Rosa26-locus expression levels, and cytosolic versus histone-fused Apex.

We thank the reviewer for raising this point. We carefully considered a cortical somatic protein preparation from the same APEX mouse line; however, there is an abundance of axonal proteins present in the cortex, which generates an important confound and could make detecting axon enriched proteins very challenging or impossible. Therefore, we decided to use H2B.APEX fusion construct. We agree that proteome differences could arise from viral vs. Rosa26-locus expression levels, or cytosolic vs. histone-fused APEX. Therefore, we include the following text to highlight the options and compromises related to different comparison samples. Lines: 203 – 207

“Thus, H2B.APEX presents one compelling option for somatic compartment control because its labeling is primarily restricted to the nucleus and soma. Although this comparison is appropriate for the study, some observed proteome differences from H2B.APEX cortex vs. Rbp4^Cre^;APEX mouse line comparison could arise from factors such as viral vs. Rosa26-locus expression levels, or cytosolic vs. histone-fused APEX.”

11. While there is a series of novelties in this work, overt "first ever" statements in the text (e.g. lines 94-96, 348-349) are redundant and inherently ambiguous in their factuality. The work's novelty is better served speaking for itself.

These statements have been removed.

12. In line 109, the authors should rephrase "revealing proline-directed kinases and phosphosites as major regulators for corticostriatal projection development" to something more closely fitting the observation that proline-directed kinases and phosphosites are dynamically regulated in corticostriatal projections during development. The current phrase infers that the functional component has been determined in this study, which would need further experimental determination (e.g. knockouts). The data provided in the manuscript only point to these proteins being highly dynamic, in both abundance and phosphorylation throughout development.

This statement has been rephrased. Lines: 107 – 108

“Combining APEX-based proximity labeling with phosphopeptide enrichment enabled a characterization of the local phosphoproteome in axons, revealing dynamic regulation of proline-directed kinases and phosphosites in corticostriatal projections during development.”

13. In line 163, authors should change the phrase "differential pattern of protein labeling" to "reduced protein abundance", as there are no obvious ratiometric changes in band intensity patterns in the western blot shown in Figure 2 supplement 1B. If the authors want to make a claim on that, they should provide a GFP blot to normalize to the amount of cells that express APEX in the region to determine if changes in protein abundance are due to the number of VGAT vs VGLUT cells, or due to differential protein abundance.

We agree that the phrase “reduced protein abundance” is more appropriate. This statement has been rephrased. Lines: 163

“Western blot analysis of total prefrontal cortex (PFC) lysates at 60 min BP incubation shows reduced protein abundance in VGAT-Cre compared to VGlut2-Cre, with negligible labeling in the Cre-negative APEX controls (Figure 2 —figure supplement 1b).”

14. The discussion requires a few additional thoughts:(i) The authors do not properly reflect the involvement of proline-directed kinases in the development of corticostriatal projections, which stands in contrast to the fact that they sell this as one of their major findings throughout the manuscript.

The following discussion is incorporated in the text. Lines: 569 –578

“Our example of phosphorylation network and correlation analysis emphasizes developmental roles of *Mapk1*, a proline-directed kinase (PDK), in the mTOR-dependent regulation of axon growth cones. Other developmentally regulated PDKs include *Cdk5* and *Gsk3b*, well-known regulators of the axon guidance pathway. Deletion of neuron-specific *Cdk5* activator protein, *p35*, disrupts callosal axons assimilation into the corpus callosum in mice and affects cortical neuron migration and axodendritic orientations (Chae et al., 1997; Kwon et al., 1999). In contrast, inactivation of *Gsk3b* promotes axon extension (Zhou et al., 2004). Deletion of *Gsk3b* in cortical neurons disrupts dendritic orientations, but spares axon elongation into the corpus callosum. Density and arborization of axons in *Gsk3b*–/– neurons remain to be investigated (Morgan-Smith et al., 2014). Altogether, proper PDK activity is essential for axonal development.”

(ii) An obvious future research target based on the present study is the role of Fyn in the development of corticostriatal connectivity. Here, a discussion of the Fyn KO phenotype might be informative (as I did not find evidence for changes in corticostriatal connectivity in the Fyn KO).

The following discussion is incorporated in the text. Lines: 579 – 593

“In this study, we identified *Fyn* as a regulator of PDKs via the kinase-substrate map. *Fyn* knockout (*Fyn* KO) shows cortical, hippocampal, and cerebellar abnormalities in neuronal migration, leading to partial Reeler phenotype and behavior (Grant et al., 1992; Kuo et al., 2005). In addition, inversion of cortical lamination was observed in *Fyn* KO mice (Yuasa et al., 2004). Therefore, we predict that dysregulated cortical stratification in *Fyn* KO mice likely affects cortical projection patterns and striatal function. *Fyn* is also known to regulate *Cdk5*, a PDK that controls actin cytoskeleton remodeling (Sasaki et al., 2002). *Fyn* KO resulted in abnormal axodendritic branching of cortical neurons due to decrease in *Cdk5* tyrosine phosphorylation and activity (Sasaki et al., 2002). The same *Cdk5* phosphosite was found in our kinase-substrate network, highlighting *Fyn* as a regulator of cell migration and subcellular morphology development. However, the specific role of *Fyn* in the development of corticostriatal connectivity is understudied, and to what extent Fyn deletion affects other PDK functions in axons remains unclear. Generation of conditional *Fyn* KO in corticostriatally projecting neurons would be informative to elucidate its role in striatal development. Altogether, our analyses show signaling crosstalk among PDKs and their regulators in developing brain.”

(iii) It is striking that a Gria1-Dlg1-Dlg4 network was discovered in the corticostriatal axon proteomics. This is unexpected – a postsynaptic network in an axonal dataset. The authors should discuss what this might mean for their new mouse line (e.g. leakiness of the APEX2 reporter, transfer of APEX2 etc.).

The following discussion is incorporated in the text. Lines: 523 – 536

“It is unexpected that postsynaptic proteins, such as the postsynaptic density 95 protein Psd95 (*Dlg4*), were detected in the axon dataset. One possibility is that such proteins exist in high abundance and are strongly associated with presynaptic partners during streptavidin enrichment, resulting in co-purification. Although unlikely, we cannot completely exclude the possibility of leakiness of the APEX2 reporter expression, or the transfer of APEX2 across compartments. Another explanation is that corticostriatal projections form axo-axonic synapses among excitatory inputs (e.g., cortical-cortical or cortico-thalamic axo-axonic synapses), since Psd95 in axons implies glutamatergic presynaptic partners. Indeed, prior ultrastructural observations identified putative glutamate-dopamine axo-axonic contacts in the striatum (Cover and Mathur, 2021). Dual color fluorescent tracing also showed rare appositions among thalamic, cortical, and basolateral amygdala inputs onto dendritic spines of spiny projection neurons in nucleus accumbens (Xia et al., 2020). Therefore, further ultrastructural studies are needed to determine whether there are glutamaterigic axo-axonic synapses on corticostriatal axons in the dorsal striatum.”

(iv) The discussion on the Netrin1-DCC pathway is not particularly strong, while aspects of the data are stronger. The authors might want to consider either getting rid of this discussion point, or putting Figure 4F in a supplemental figure. The data provided only show phosphorylation events, not activity of a specific pathway, as Fyn is involved in a lot of different processes. Figure 4 supplement 1B may be a more important panel to have in Figure 4.

The original Figure 4f is now Figure 4 – supplement 2.